# Deubiquitinase activity is required for the proteasomal degradation of misfolded cytosolic proteins upon heat-stress

Nancy N. Fang[1], Mang Zhu[1], Amalia Rose[1], Kuen-Phon Wu[2] & Thibault Mayor[1]

Elimination of misfolded proteins is crucial for proteostasis and to prevent proteinopathies. Nedd4/Rsp5 emerged as a major E3-ligase involved in multiple quality control pathways that target misfolded plasma membrane proteins, aggregated polypeptides and cytosolic heat-induced misfolded proteins for degradation. It remained unclear how in one case cytosolic heat-induced Rsp5 substrates are destined for proteasomal degradation, whereas other Rsp5 quality control substrates are otherwise directed to lysosomal degradation. Here we find that Ubp2 and Ubp3 deubiquitinases are required for the proteasomal degradation of cytosolic misfolded proteins targeted by Rsp5 after heat-shock (HS). The two deubiquitinases associate more with Rsp5 upon heat-stress to prevent the assembly of K63-linked ubiquitin on Rsp5 heat-induced substrates. This activity was required to promote the K48-mediated proteasomal degradation of Rsp5 HS-induced substrates. Our results indicate that ubiquitin chain editing is key to the cytosolic protein quality control under stress conditions.

[1] Department of Biochemistry and Molecular Biology, Michael Smith Laboratories, University of British Columbia, 2125 East Mall, Vancouver, British Columbia, Canada V6T1Z4. [2] Department of Structural Biology, St Jude Children's Research Hospital, Memphis, Tennessee 38105, USA. Correspondence and requests for materials should be addressed to T.M. (email: mayor@mail.ubc.ca).

Intricate protein quality control (PQC) degradation pathways have evolved in eukaryotic cells to eliminate misfolded polypeptides and maintain protein homeostasis[1–3]. The accumulation of misfolded proteins and their aggregation have been implicated in numerous proteinopathies, like Parkinson's and Alzheimer's diseases[1,4]. A large portion of misfolded proteins are degraded by the ubiquitin proteasome system, which relies on a cascade of enzymes (E1 ubiquitin-activating enzyme, E2 ubiquitin-conjugating enzyme and E3 ubiquitin ligase) that first poly-ubiquitinate targeted proteins before their proteolysis by the proteasome[5,6]. Several compartmentalized degradation PQC pathways have been identified, in which E3 ubiquitin ligases selectively ubiquitinate misfolded polypeptides for proteasomal degradation, often with the help of chaperone proteins to mediate substrate recognition[2,7,8]. A major challenge is to elucidate how the cell makes the triage decision between folding and degradation in the cytosol. As well, as many cytosolic misfolded proteins are also degraded by the lysosomes via autophagy, it is unclear how a specific proteolytic route is selected for a given PQC target.

Heat-shock (HS) elicits a complex cellular response in which the folding capacity of the cell is elevated to alleviate protein misfolding[9,10], while ubiquitination levels and proteasome degradation are increased[11,12]. Hul5 and Rsp5 are the two main ubiquitin ligases responsible for the rapid increase in poly-ubiquitination levels and proteasomal degradation of misfolded proteins upon HS in yeast cells[13,14]. Hul5 ligase mainly targets low solubility cytosolic proteins in both unstressed conditions and after HS[13]. Hul5 is associated to the proteasome[15], and its closest human homologue Ube3C was also shown to increase proteasome processivity to promote degradation of misfolded proteins[16]. Owing to Hul5 E4 activity that elongates ubiquitin chains[15], we proposed that this ubiquitin ligase could function at the proteasome to enhance proteolytic signals primed by other E3s on misfolded proteins[17]. Rsp5, on the other hand, employs a bipartite substrate recognition mechanism that is based on (1) the interaction with the Hsp40 co-chaperone Ydj1, which presumably acts as substrate adaptor protein, and (2) heat-exposed PY motifs on misfolded substrates which can be recognized directly by Rsp5 (ref. 14). Underlying the importance of this Rsp5 pathway, downregulation of its closest mammalian homologue Nedd4 also led to an impairment of the HS-induced increased poly-ubiquitination[14]. Rsp5 emerges as a key ubiquitin ligase with a major role in maintaining protein homoeostasis as it has been implicated in the nuclear export of mRNAs key to the HS response[18,19], the lysosomal degradation of misfolded plasma membrane proteins and aggregation-prone cytosolic proteins[20,21]. In mammalian cells, Nedd4 has also been shown to promote degradation of α-synuclein that is involved in Parkinson's disease[22], and the NAB compound that targets the Rsp5/Nedd4 pathway was shown to reduce α-synuclein toxicity[23]. Rsp5 has previously been shown to catalyse mono- or K63-linked ubiquitination to mediate endocytosis and the synthesis of unsaturated fatty acids and sterols[19,24–27]. In agreement, in vitro experiments also confirmed that Rsp5 activity is more specific to K63- than K48-ubiquitin chains[28]. Intriguingly, Rsp5 is required to mediate the buildup of K48-linked poly-ubiquitin chains after HS[14], consistent with its role in targeting these misfolded substrates to the proteasome[29]. It remained unclear how Rsp5 can promote the conjugation of non-K63 linkages, such as K48 chains, for the proteasomal degradation of cytosolic misfolded substrates upon HS.

Ubiquitination is highly dynamic and reversible due to the action of deubiquitinases that are specific ubiquitin proteases. Deubiquitinases not only generate a pool of free ubiquitin in the cell but also regulate many cellular processes by fine-tuning or modulating substrate ubiquitination[30,31]. In many cases, deubiquitinases have been shown to antagonize the activity of ubiquitin ligases to prevent proteasomal degradation of their substrates. For instance, the deubiquitinase Ataxin-3 was shown to interact with C-terminus of Hsc70 Interacting Protein (CHIP), a major mammalian E3 ligase involved in cytosolic PQC, to limit the length of ubiquitin chains built on CHIP substrates[32]. Deubiquitinases can also discriminate ER-associated proteins that should or should not be degraded by the proteasome[33]. Similarly, proteasomal deubiquitinases, such as Ubp6 in yeast and Usp14 in mammals, can limit proteolysis of some proteasome substrates[15,33], although the role of the proteasomal deubiquitinases is primarily thought to be required for processing substrates before their degradation[34–37].

In this study, we discovered that yeast Ubp2 and Ubp3 are two deubiquitinases that promote proteasomal degradation of cytosolic misfolded proteins targeted by Rsp5 after heat stress. We found that these deubiquitinases prevent the assembly of K63-linked ubiquitin chains to promote the buildup of K48-linked ubiquitin chains by Rsp5, which displays altered linkage specificity under stress conditions.

## Results

**Ubiquitination levels after HS are limited by *UBP2* and *UBP3*.**
Degradation of cytosolic misfolded proteins is likely controlled by both the action of PQC E3 ligases and antagonizing deubiquitinases, which could prevent the proteolysis of transiently misfolded proteins. To identify a deubiquitinase that could potentially regulate the degradation of cytosolic PQC targets, we assessed the cellular ubiquitination levels after a short and acute HS in a panel of *S. cerevisiae* mutant strains deleted for each of the non-essential deubiquitinases (19 in total). We previously showed that mostly low solubility cytosolic proteins were further ubiquitinated in similar conditions[14,38]. We first compared ubiquitination levels in HS-treated cells incubated at 45 °C for 5 min, with cells maintained at 25 °C, using a dot-blot assay (Fig. 1a). In this case, we opted for a shorter HS period in comparison with our previous work (15 min) to notably avoid a possible faster depletion of free mono ubiquitin in the absence of a deubiquitinase. We found that ubiquitination levels after HS were markedly elevated in *ubp2Δ* and *ubp3Δ* cells compared with wild-type (WT) cells. We confirmed the pronounced differences in independent experiments (Fig. 1b, c and Supplementary Fig. 1a). Ubiquitination levels were also elevated in *ubp7Δ* and *10Δ*, but to a lesser extent (Fig. 1a), and the differences were not significant (Supplementary Fig. 1b). Similar levels of ubiquitination were also quantified in *ubp2Δ* and *ubp3Δ* cells by conventional western blot analysis, and the higher conjugation levels were persistent after prolonged HS (Supplementary Fig.1c,d). Importantly, while plasmid-expressed WT deubiquitinases restored normal ubiquitination levels in the mutant cells, expression of catalytically inactive mutants did not (Fig. 1d,e). These results indicate that *UBP2* and *UBP3* are the two main deubiquitinases in yeast that control ubiquitination levels after HS. Overall ubiquitination levels after HS were more elevated in *ubp2Δ* than *ubp3Δ* cells, and raised further in the absence of both deubiquitinases (Fig. 1f). As well, deletion of *UBP2* led to higher conjugation levels in both mild and acute heat-stress conditions, whereas elevated ubiquitination levels in *ubp3Δ* cells were only observed at higher temperatures (Supplementary Fig. 1e). These data suggest that Ubp2 may have a more prevalent role in preventing elevated ubiquitination levels following HS, whereas Ubp3 may become more important upon the increased load of misfolded proteins.

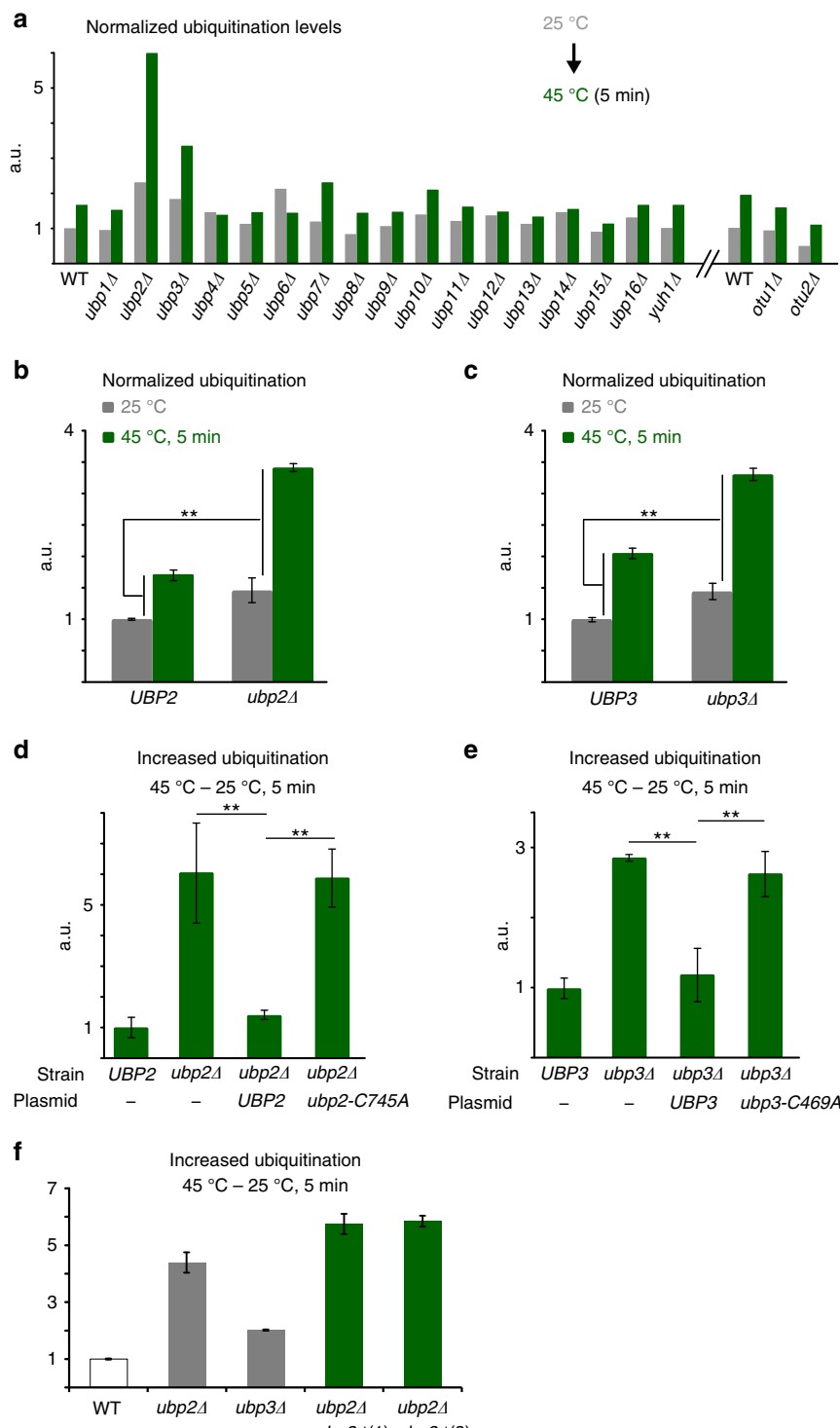

**Figure 1 | Ubp2 and 3 control ubiquitination levels upon HS.** (**a**) Ubiquitination levels were quantified by dot-blots and normalized to Pgk1 levels in WT and the indicated single-deletion strains. Cells were maintained at 25 °C or incubated at 45 °C for 5 min. (**b,c**) Normalized ubiquitination levels were quantified by dot-blots (shown in Supplementary Fig. 1A) as in A and compared with a two-tail student *t*-test (\*\*$P < 0.01$). (**d,e**) Differences of normalized ubiquitination levels between unstressed cells (25 °C) and HS cells (45 °C for 5 min) quantified by dot-blots in the designated strains with the indicated plasmids ('-' denotes control empty plasmid). Increased ubiquitination levels were compared using a two-tail student *t*-test (\*\*$P < 0.01$). (**f**) Differences of normalized ubiquitination levels before and after HS (45 °C for 5 min) quantified by dot-blots in the indicated cells. All experiments in **b**–**f** were done with three biological replicates and averaged values are shown with s.d. Results are shown as arbitrary units (a.u.) and each value is relative to the reference sample.

**Ubp2 and 3 antagonize Rsp5-dependent ubiquitination after HS.** We next sought to determine which E3 contributed to the increased ubiquitination levels in absence of the two deubiquitinases. We first assessed whether the lack of Hul5 or Rsp5 activity could mitigate the phenotype, as both ubiquitin ligases mostly contribute to the HS-induced increased ubiquitination[13,14]. We found that the elevated ubiquitination levels in *ubp2Δ* cells after HS was mostly abrogated in *rsp5-1* cells, while only marginally

(but significantly) affected in *hul5Δ* cells (Fig. 2a). Similar results were observed when assessing *ubp3Δ* with *rsp5-1* and *hul5Δ* alleles (Fig. 2b). Our data suggest that Ubp2 and Ubp3 predominantly deubiquitinate Rsp5 substrates after HS, although both deubiquitinases could also in some cases cleave off moieties added by other ligases such as Hul5.

Consistent with a model where Ubp2 and Ubp3 antagonize Rsp5 activity after HS, both deubiquitinases were previously shown to function with Rsp5. Ubp2, with the help of the adaptor protein Rup1, forms a complex with Rsp5 to regulate biogenesis and sorting of multivesicular bodies and RNA polymerase II degradation[39–42]. Ubp3, with its co-activator Bre5, mediates deubiquitination of the Rsp5 substrates Sec23, a component of the COPII complex, and the TFIID transcription factor[43,44]. To verify the involvement of the two deubiquitinases, we therefore investigated whether *RUP1* and *BRE5* were also required to control ubiquitination levels upon HS. Accordingly, ubiquitination levels in *rup1Δ* and *bre5Δ* cells after HS were similarly elevated as in *ubp2Δ* and *ubp3Δ* cells, respectively (Supplementary Fig. 1f).

To further demonstrate the role of Ubp2 and Ubp3 in the Rsp5-dependent pathway for targeting heat-induced misfolded proteins, we next assessed the physical association of Ubp2 and Ubp3 with Rsp5 in HS stress conditions. We crosslinked cells *in vivo* before lysis to preserve possible heat-induced interactions during the immunoprecipitation (IP) of the TAP-tagged Ubp2 and Ubp3. Although less proteins were soluble and immunoprecipitated after HS in these conditions, we found that higher levels of Rsp5 were recovered by co-IP of both Ubp2 and Ubp3 after HS (Fig. 2c,d), whereas the unrelated Pgk1 protein, which was used as a loading control, was not readily detected. Pulldown of Rsp5 with Ubp2 in the HS conditions was specific because it required *RUP1* (Supplementary Fig. 2a). We also observed an increased interaction of the deubiquitinases with Rsp5 after HS in the reciprocal experiments (Supplementary Fig. 2b,c) and verified that both deubiquitinases were not degraded after HS (Supplementary Fig. 2d). These results suggest that more Ubp2 and Ubp3 interact with Rsp5 upon heat stress to process Rsp5 substrates.

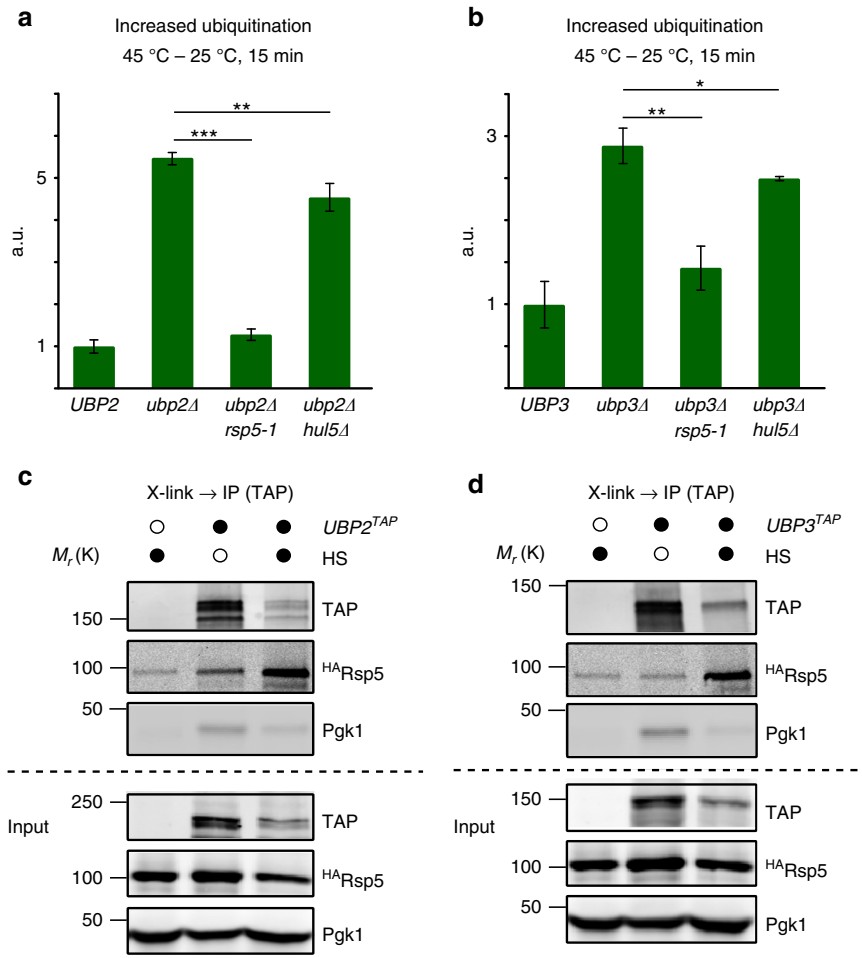

**Figure 2 | Ubp2 and 3 mainly deubiquitinate Rsp5 substrates and have increased interactions with Rsp5 after HS.** (**a,b**) Increased ubiquitination levels after a 15 min HS at 45 °C in the indicated cells quantified by dot-blot and compared with unstressed cells (25 °C). Experiments were done with three biological replicates and averaged values are shown with s.d. In each experiment, increased ubiquitination levels were compared, as shown, to the deletion or WT strains using a two-tail student *t*-test (***$P < 0.001$; **$P < 0.01$; *$P < 0.05$). a.u., arbitrary units. (**c,d**) Cells expressed 3xHA-tagged Rsp5 from a plasmid and endogenously C-terminally TAP-tagged *UBP2* or *UBP3*, when indicated (black circles). Before lysis, cells that remained at 25 °C (unfilled circles) or heat shocked (40 °C, 20 min; black circles) were crosslinked for the remaining 10 min with 1% formaldehyde. Western blots of the indicated proteins are shown.

**HS-induced degradation requires *UBP2* and *UBP3*.** We first reasoned that deletion of *UBP2* or *UBP3* could lead to an accelerated degradation of HS-induced Rsp5 substrates, based on the observed increased ubiquitination in these mutant cells. We previously showed that Rsp5 mediates the proteasomal degradation of short-lived proteins misfolded by elevated temperatures[14]. The degradation of these heat-induced proteins by the proteasome was first characterized by Goldberg and colleagues, and was shown to occur after shifting cells to 38 °C (ref. 12). To our surprise, deletion of either *UBP2* or *UBP3* abrogated the heat-induced degradation of proteins pulsed-labelled with [35]S (Fig. 3a,b). Similarly, we found that turnover of the cytosolic model substrate Cdc19-D367R[MYC], which is degraded by the proteasome at higher temperatures in an Rsp5-dependent manner due to the destabilizing point mutation[14], was inhibited in *ubp2Δ* and *ubp3Δ* cells (Fig. 3c). All together these results indicate that both deubiquitinases have an unexpected role in promoting proteasomal degradation of cytosolic misfolded targeted by Rsp5 upon HS.

**HS-induced ubiquitinated proteins in absence of Ubp2 or Ubp3.** One possible explanation for our observations is that the deletion of the deubiquitinases could lead to a shift in Rsp5-substrate specificity, for example, by allowing more ubiquitination of membrane anchored proteins (that would then not be efficiently degraded by the proteasome). To test this possibility, we used a triple-stable isotope labeling with amino acid in cell culture (SILAC) mass spectrometry approach combined with the antibody-based enrichment of peptides with diGly, the remnant of conjugated ubiquitin after tryptic digest[45]. We compared diGly site occupancy of conjugated proteins in heat-stressed *ubp2Δ* or *ubp3Δ* cells (light-labelled) to WT cells with (medium-labelled) or without (heavy-labelled) HS treatment (Fig. 3d). Cells were heat shocked for 20 min, which led to an increase (over twofold) of ~70% of the quantified diGly sites in the WT cells (283/398 and 336/466; Fig. 3e,f and Supplementary Data 1). In contrast, only ~5 and 10% of the diGly sites were further increased over twofold in *ubp2Δ* and *ubp3Δ* cells, respectively (Fig. 3e,f). These results indicate that the absence of *UBP2* or *UBP3* does not lead to the ubiquitination of a vastly different pool of heat-induced substrates nor prevent the conjugation of alternative sites within a given substrate.

To better evaluate which proteins were affected by the absence of either deubiquitinases, we performed an additional experiment using a shorter HS treatment. We reasoned that a greater difference to WT cells could then be observed. In this triple-SILAC experiment, we assessed diGly site occupancy in *ubp2Δ* (light-labelled), WT (medium-labelled) and *ubp3Δ* (heavy-labelled) cells heat shocked for 5 min (Supplementary Fig. 3a). Under these conditions, and in contrast to the longer HS treatment, a large fraction of diGly sites (161/231) were further enriched when derived from either *ubp2Δ* or *ubp3Δ* cells (Supplementary Fig. 3b and Supplementary Data 1). Interestingly, these sites were similarly affected by the absence of either *UBP2* or *UBP3*, indicating that both deubiquitinases mostly process the same pool of Rsp5 substrates. In agreement with our previous study, proteins affected by the absence of either deubiquitinase were mostly cytosolic (Supplementary Fig. 3c). However, among the proteins only found ubiquitinated in absence of Ubp2 (46%) or Ubp3 (43%) after 5 min HS (but not after 20 min HS), a larger fraction consisted of membrane-associated proteins (Supplementary Fig. 3d). One possibility is that these proteins correspond to Rsp5 substrates that were further ubiquitinated before the HS, indicating that the absence of one of the two deubiquitinases may also slightly alter composition of conjugated

proteins after a short HS exposure. Nevertheless, our data show that there are few changes in site occupancies between WT and *ubp2Δ* or *ubp3Δ* cells after longer HS (that is, 15–20 min), while we observed increased ubiquitination levels after similar HS treatments (Fig. 2a,b and Supplementary Fig. 1e). Taken together, these results suggest that the absence of either deubiquitinase likely leads to the attachment of longer ubiquitin chains on cytosolic heat-induced Rsp5 substrates.

**Ubp2 and -3 suppress conjugation of K63-linked chains upon HS.** We next hypothesized that Ubp2 and Ubp3 could affect the composition of ubiquitin chains to promote proteasomal degradation of the heat-induced cytosolic misfolded proteins. While Rsp5 preferably synthesizes K63-linked chains, which generally do not support proteasomal degradation in the cell[46], the absence of Rsp5 affected the accumulation of K48-linkages after HS[14,47] (see also Supplementary Fig. 4a). As Ubp2 can specifically cleave off K63- but not K48-linked ubiquitin *in vitro*[41], the two deubiquitinases could prevent the assembly of K63 chains on heat-induced Rsp5 substrates.

To test the above possibility, we first assessed which types of ubiquitin chains accumulate in absence of the deubiquitinases using antibodies specific to K48- or K63-linked chains. While levels of K48-linked chains increased after HS in WT cells, their levels did not markedly increase further in *ubp2Δ* or *ubp3Δ* cells after HS (Fig. 4a). In contrast, K63-linked chains were distinctly elevated after HS in *ubp2Δ* and *ubp3Δ* cells in comparison with WT cells (Fig. 4a). We verified the specificity of the two antibodies with commercial K48 and K63 poly-ubiquitin chains (Supplementary Fig. 4b). We also obtained similar data when we expressed Myc-tagged ubiquitin mutants containing only K48 or K63 (while the other six K were mutated to R). Levels of poly-ubiquitin chains containing the K63-only mutant were elevated after HS in *ubp2Δ* and *ubp3Δ* cells in comparison with WT cells (2.6- and 2.4-fold, respectively), whereas levels of chains containing the K48-only mutant were not noticeably altered (Fig. 4b). The higher levels of chains containing the K63-only mutant were also observed both after short (5 min) and prolonged HS (10–15 min; Supplementary Fig. 4c).

To independently confirm our results, we used a selected reaction monitoring (SRM) approach to quantify cellular levels of K48 and K63 chains by mass spectrometry in both WT and *ubp2Δ* cells. We first enriched, after HS, proteins conjugated to $H_8$-ubiquitin (with an N-terminal octohistidine tag) by nickel chromatography, and then isolated high molecular weight species on a protein gel. After spiking the labelled synthetic ubiquitin peptides into each sample to quantify changes of both linkages, we found that there was a significantly higher increase of K63-linked chains than K48-linkages upon deletion of *UBP2* (Fig. 4c). We next directly verified these observations on three Rsp5 substrates: Sup45, Pdc1 and Cdc19 (ref. 14). All three substrates displayed higher levels of poly-ubiquitination in the absence of *UBP2* (Supplementary Fig. 4d). Using SRM, we confirmed that K63- but not K48-linked ubiquitin was further conjugated to these three immunoprecipitated HA-tagged substrates in *ubp2Δ* cells after HS (Fig. 4d).

**Rsp5 associates to K48-linked chains upon HS *in vivo*.** While we found that both Ubp2 and Ubp3 restrict K63-linked conjugation on heat-induced Rsp5 substrates, it remained to be determined which ligase catalyses K48-linked chains. One possibility is that another E4 ligase elongates chains primed by Rsp5 (Fig. 5a). For instance, the Elc1/Cul3 ligase promotes the degradation of RNA polymerase II that is first ubiquitinated by Rsp5 (ref. 42). However, deletion of either *ELC1* or *CUL3* did not impair

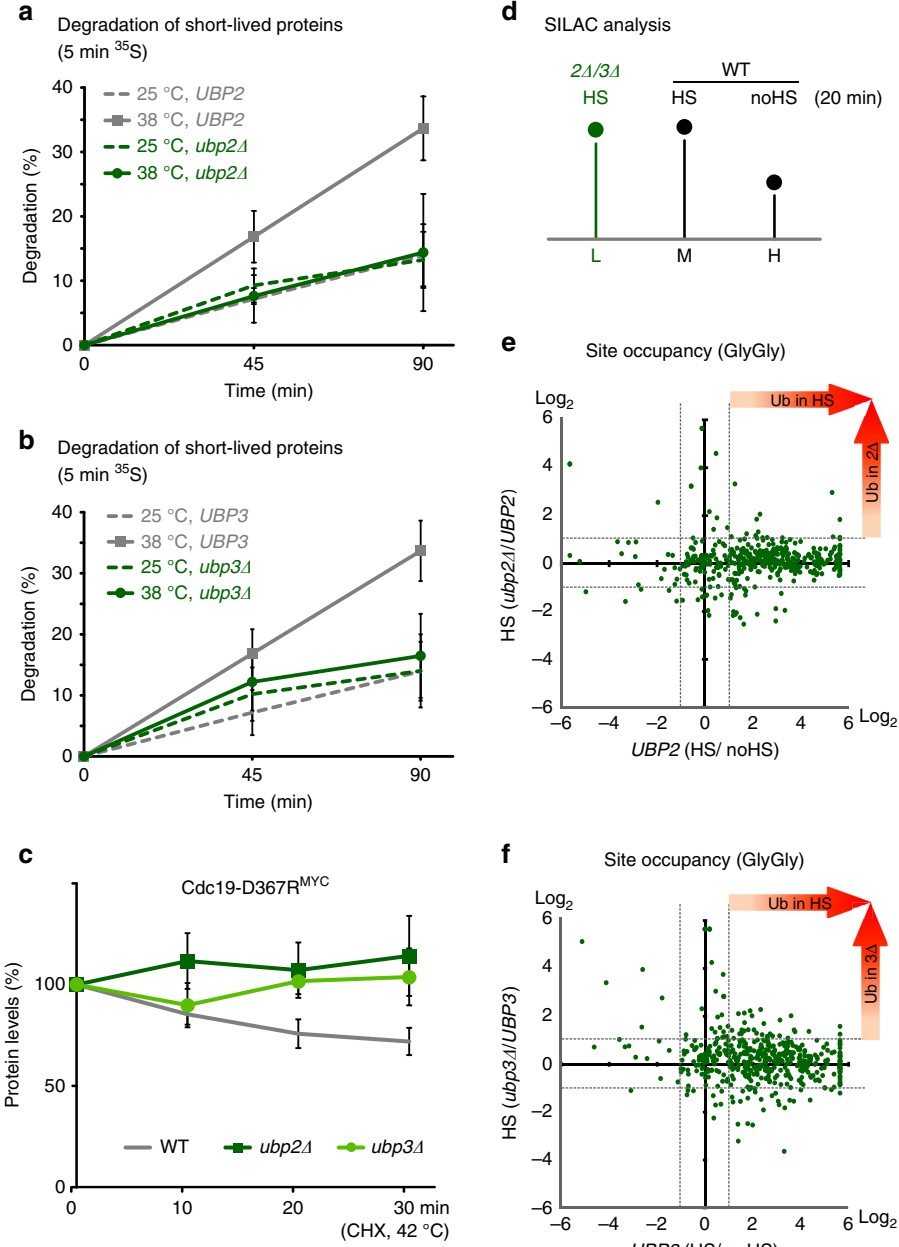

**Figure 3 | Ubp2 and 3 share substrate pools and are required for the degradation of misfolded proteins upon HS.** (**a,b**) Degradation of [35]S pulsed-labelled proteins (5 min) in WT (grey) and indicated deletion strains (green) cells at 25 °C (dotted lines) or 38 °C (straight lines). The portion of proteins degraded at the indicated times was reported from three independent experiments with s.d. (**c**) Turnover of plasmid-expressed Cdc19-D367R[13MYC] in designated cells at 42 °C was assessed by western blots after the addition of 100 μg ml[−1] cycloheximide. The Pgk1-normalized protein levels were averaged from three independent experiments and shown with s.d. (**d**) Schematic representation of the triple-SILAC experiment to quantify conjugated substrate peptides. L, M, and H denote the three different SILAC-labels (light, medium and heavy). (**e,f**) Scatter plots of the log₂ ratios of quantified conjugated peptides. In the x axis, ratios of M versus H are shown to display sites affected by heat-stress in WT cells. In the y axis, ratios of L versus M are reported to display sites affected by the absence of *UBP2* (**e**) or *UBP3* (**f**) upon HS.

the increased ubiquitination after HS (Supplementary Fig. 5a). Deletion of the *UFD2* gene that encodes for an E4 also did not affect HS-induced increased ubiquitination[13]. As well, Hul5 promotes the conjugation of K63- and not K48-linked chains after HS[14].

To determine whether Rsp5 itself could assemble K48-linked chains on its heat-induced substrates *in vivo*, we assessed which ubiquitin linkages were conjugated to proteins that co-immuno-precipitated with the ligase after crosslinking cells. We co-expressed [HA]Rsp5 and Myc-tagged ubiquitin mutants

containing only K48 or K63 (while other K were mutated to R) in cells that were or not subjected to HS. We found that Rsp5 was mostly associated to K48-only ubiquitin after HS, while K63-only ubiquitin was dominantly associated to Rsp5 in unstressed conditions (Supplementary Fig. 5b). We verified that the co-IP of conjugates was dependent on protein crosslinking and the expression of catalytically active Rsp5 (Supplementary Fig. 5c). As well, K63- but not K48-linked conjugates were co-immuno-precipitated with Hul5 in these conditions (Supplementary Fig. 5d), in agreement with our previous findings[14]. These

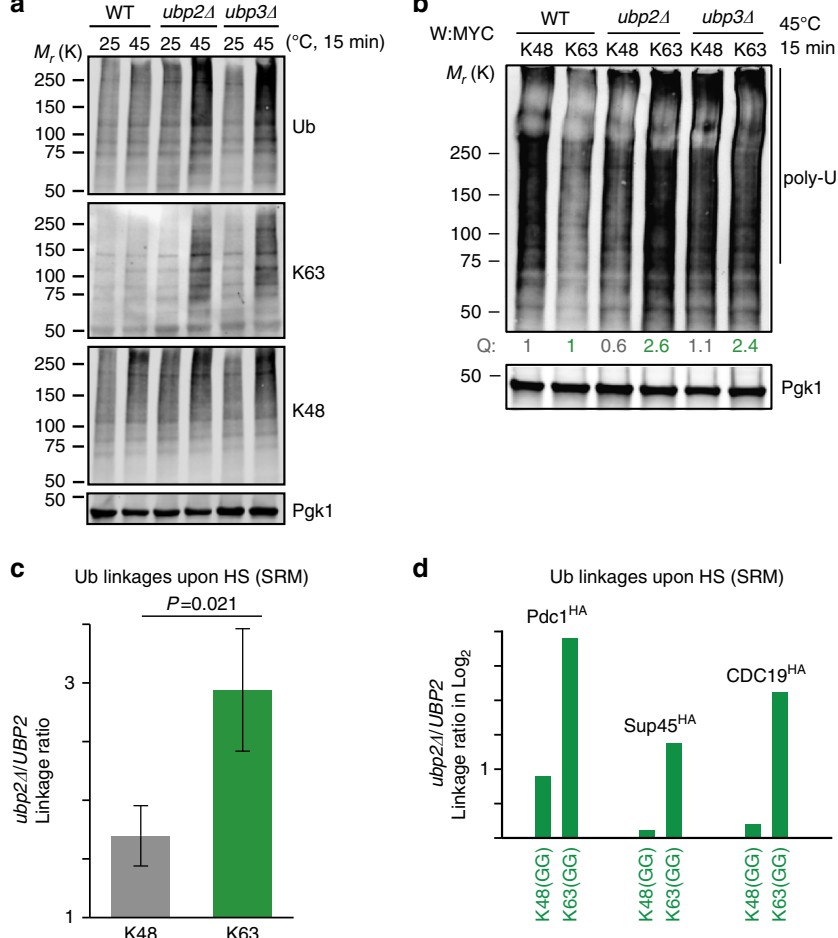

**Figure 4 | Ubp2 and 3 suppress the buildup of K63-linked chains after HS *in vivo*.** (**a**) Levels of K48- and K63-linked ubiquitin in designated cells were assessed using the chain-specific antibodies in cells grown at 25 °C and HS at 45 °C for 15 min. (**b**) Indicated cells that expressed $^{MYC}$ubiquitin-K48 only (K48) or $^{MYC}$ubiquitin-K63 only (K63) were incubated at 45 °C for 15 min. Quantified values (Q) of anti-Myc signal (>75 kDa) normalized to Pgk1 are shown relative to WT cells. (**c**) Levels of K48- and K63-linked ubiquitin in *ubp2Δ* and WT cells that expressed H$_8$-ubiquitin under the *GPD1* promoter were quantified by SRM after Ni$^{2+}$ chromatography and SDS–PAGE separation. Cells were heat shocked at 45 °C for 20 min and ratios of levels between *ubp2Δ* and WT were measured in three independent experiments (with s.d.) using the indicated spiked-in labelled diGly peptides and compared with a two-tail student *t*-test. (**d**) The levels of K48- and K63-linked ubiquitin conjugated to the indicated Rsp5 substrates were quantified by SRM. After HS (45 °C for 20 min) endogenously C-terminally 3 × HA-tagged substrates were purified using the anti-HA antibody and separated by SDS–PAGE before in-gel trypsin digest of the high molecular weight species. Ratios of the indicated ubiquitin linkages between *ubp2Δ* and WT cells are shown.

control experiments indicate that the conjugated proteins that co-immunoprecipitated with Rsp5 after crosslinking were most likely substrates of the ligase. These experiments suggest that after HS, Rsp5 associates prominently with proteins conjugated to K48 poly-ubiquitin chains. To determine whether the K48 chains conjugated to the co-immunoprecipitated proteins were conjugated by Rsp5, and not another ligase, we repeated the experiment using the I537D mutant that predominantly impairs Rsp5's ability to buildup poly-ubiquitin chains while affecting less substrate mono-ubiquitination[48]. Strikingly, less K48-only ubiquitin was bound to Rsp5-I537D compared with WT Rsp5 after crosslinking heat shocked cells (Fig. 5b and Supplementary Fig. 5c,e). Furthermore, we confirmed that mostly poly-ubiquitination—and not mono-ubiquitination—of the Cdc19 heat-induced substrate was affected in cells expressing Rsp5-I537D (Supplementary Fig. 5f). Therefore, these results suggest that Rsp5, and not another E4, plays a direct role in building up K48-linked chains on its heat-induced targets (Fig. 5a). Importantly, association to K48-only ubiquitin with Rsp5 after HS was dependent on the presence of *UBP2* and *UBP3*, as Rsp5

was mostly associated to K63-only ubiquitin in *ubp2Δ* and *ubp3Δ* cells (Fig. 5c,d). The association of Rsp5 to the K48-linked substrates was restored upon the plasmid-driven expression of the WT but not catalytically inactive deubiquitinases (Fig. 5e,f). These results indicate that, under heat stress conditions, Rsp5 most likely specifically assembles K48-linked ubiquitin chains on its substrates with the help of both Ubp2 and Ubp3.

**Rsp5 can assemble K48 chains *in vitro*.** One possibility is that Rsp5 linkage specificity could be altered at higher temperatures to promote the assembly of K48-linked chains. We first verified that Rsp5 was active at higher temperatures *in vitro* by assessing its activity in the absence of any substrate (Supplementary Fig. 6a). Using single lysine ubiquitin mutants, we observed that more K63 chains were assembled at 30 °C in comparison with K48 chains (Fig. 6a and Supplementary Fig. 6b), in agreement with previous work. However, both linkages accumulated to the same levels at higher temperatures (Fig. 6a and Supplementary Fig. 6b). There was no increase in conjugation at the higher temperature when

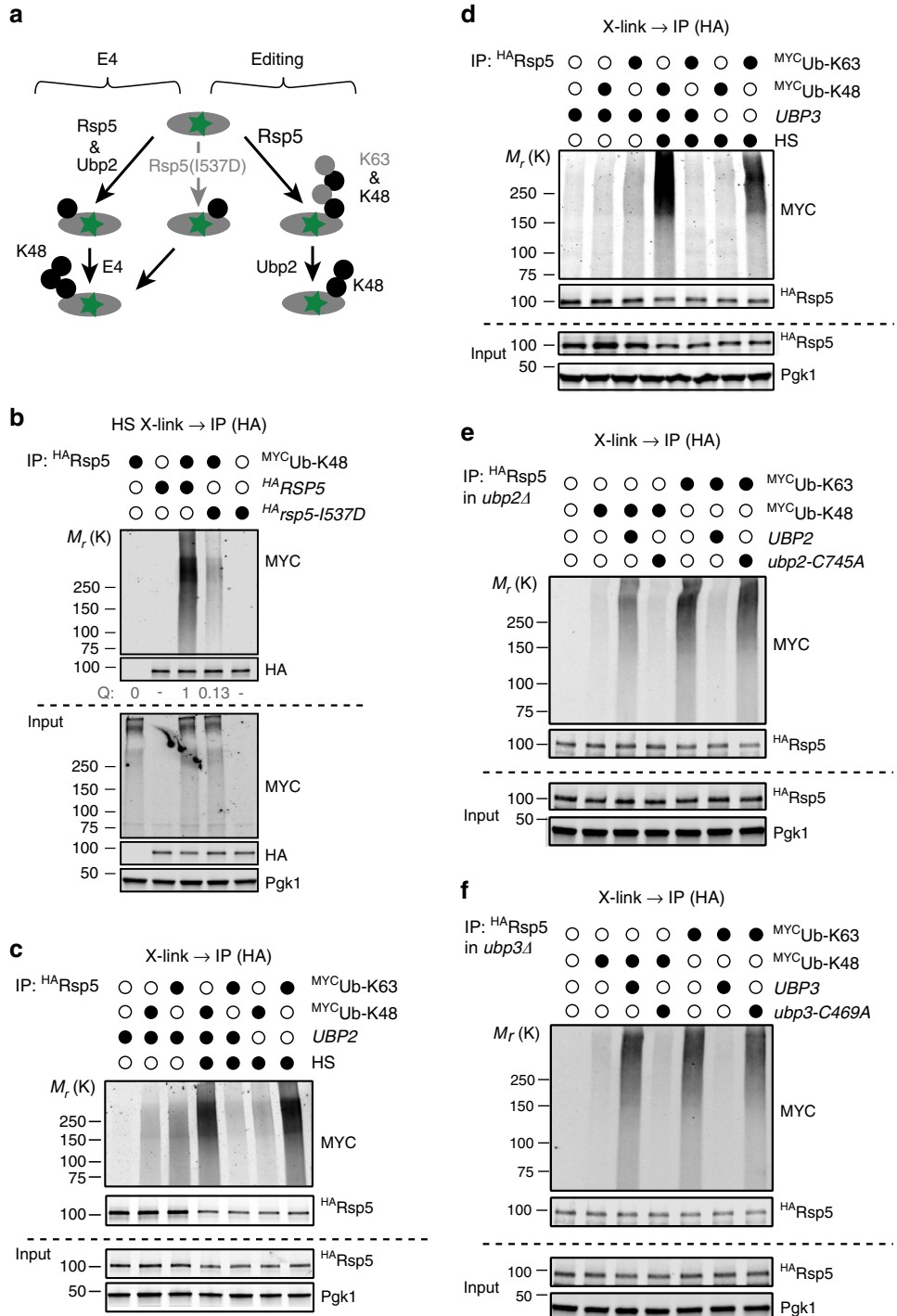

**Figure 5 | Rsp5 conjugates K48-linked ubiquitin chains to its heat-induced substrates *in vivo* in an Ubp2 and 3-dependent manner.** (**a**) Schematic representation of two possible models to explain the role of Rsp5 in the assembly of K48 chains after HS. The I537D mutant of Rsp5 that impairs mostly poly- but not monoubiquitination should only strongly affect the editing pathway. (**b**) Cells that expressed ³HARsp5 or ³HARsp5-I537D together with empty or MYCubiquitin-K48 only constructs were heat shocked for 20 min (40 °C) and crosslinked with 1% formaldehyde for the remaining 10 min before lysis with SDS and the anti-HA IP. Western blot analysis of both IP and input samples are shown. (**c,d**) WT and *ubp2Δ* (**c**) or *ubp3Δ* (**d**) cells that expressed ³HARsp5 together with empty, MYCubiquitin-K48 only or MYCubiquitin-K63 only constructs were heat shocked (40 °C, 20 min) or not, and crosslinked with 1% formaldehyde before lysis with SDS and IP with anti-HA antibody-conjugated magnetic beads. Western blot analysis of both IP and Input samples are shown. (**e,f**) As in **b,c**, the *ubp2Δ* (**e**) or *ubp3Δ* (**f**) cells carried out either an empty control plasmid or a plasmid that expressed the WT or catalytically inactive deubiquitinases.

we employed a lysine-less (K0) ubiquitin mutant (Supplementary Fig. 6c), indicating that mostly poly-ubiquitin chains were assembled in these conditions. The change of K63 versus K48

ratios at higher temperature was specific to Rsp5, as it was not observed with the Trim5 RING E3 domain (Supplementary Fig. 6d), which preferably assembles K63-linked chains[49]. We

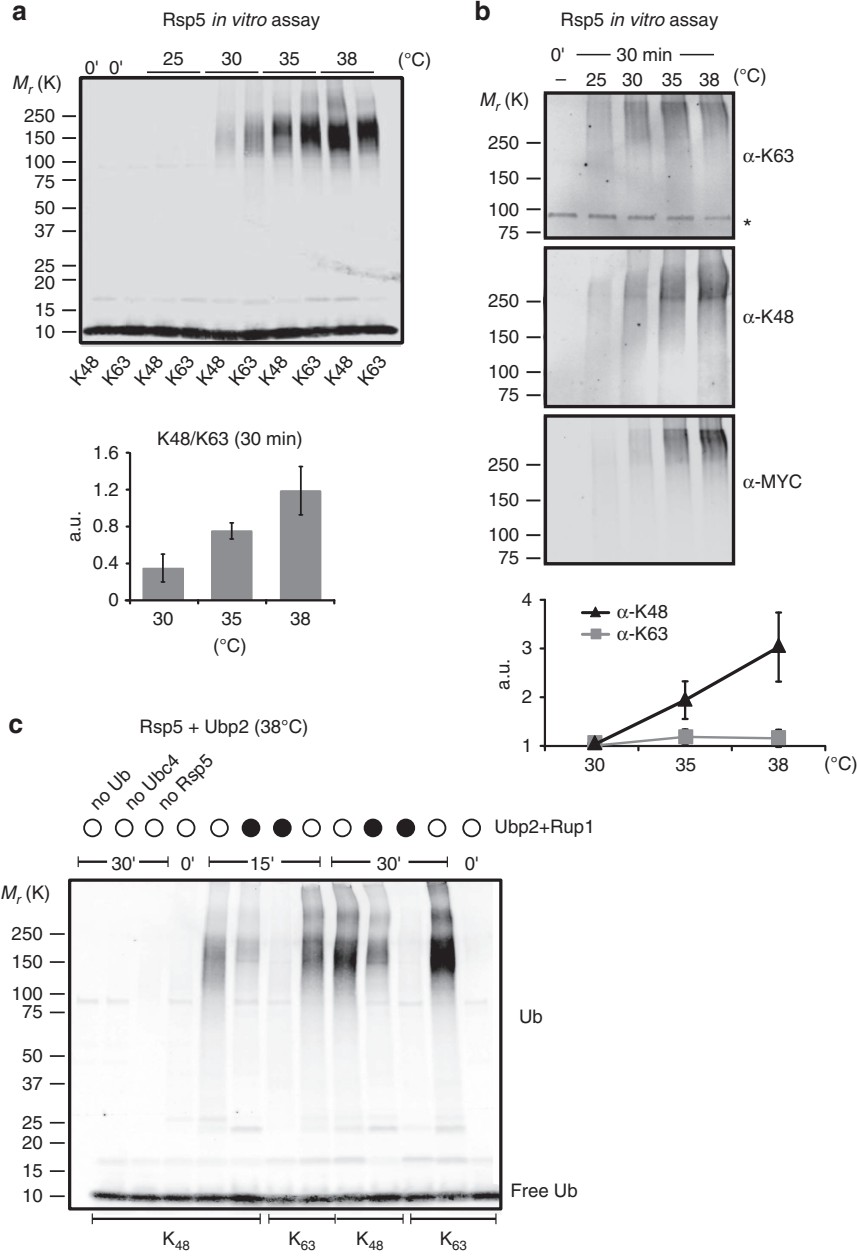

**Figure 6 | Rsp5 specificity is altered at higher temperatures and promotes K48 chains assembly in the presence of Ubp2 *in vitro*.** (**a**) *In vitro* Rsp5 auto-ubiquitination assay for 30 min at the indicated temperatures with His$_6$-tagged K48-only (K48) or K63-only (K63) ubiquitin variants, followed by anti-ubiquitin western blots. Histograms (right) show quantification of K48 versus K63 levels (>100 kDa) from three independent experiments (with s.d.). Levels at 25 °C were too close to the background for quantification. (**b**) *In vitro* Rsp5 auto-ubiquitination assay at the indicated temperatures for 30 min with the Myc-tagged ubiquitin followed by western blots with anti- K63 chains, K48 chains and Myc antibodies. Asterisk denotes an unspecific band. Relative levels were quantified in three independent experiments (with s.d.). (**c**) *In vitro* Rsp5 auto-ubiquitination assay at 38 °C incubated for 15 or 30 min followed by western blot with anti-ubiquitin antibody. Unless indicated otherwise, the reactions were done in the presence of E1, Ubc4, Rsp5, His$_6$-ubiquitin and an equimolar amount of Ubp2 and Rup1 (black circles).

then confirmed that the levels of K48-linked chains markedly increased at the higher temperature in this assay using WT MYC-tagged ubiquitin and linkage-specific antibodies, whereas levels of K63-linked ubiquitin remained stable (Fig. 6b).

As Ubp2 is more specific for K63-linked ubiquitin than K48 (ref. 41), the combined action of the deubiquitinases and the heat-induced altered chain specificity of Rsp5 could promote the assembly of K48 chains in stressed conditions. To test this idea, we first performed an *in vitro* ubiquitination reaction with Rsp5 together with Ubp2 and the Rup1 adaptor protein using K48-only

or K63-only ubiquitin mutants. We found that K48-linked chains gradually accumulated in the presence of the deubiquitinase (Fig. 6c). In contrast, K63 chains were only assembled in the absence of the deubiquitinase.

The aforementioned ubiquitination reactions were done in absence of substrates. It should be noted that we were not able to verify the same phenomena (that is, increase conjugation of K48 chains) when we employed a His$_6$-tagged fragment of Wbp2 (169–256) that contains a single lysine residue (K222; Supplementary Fig. 6e). Wbp2 is a human protein that has been

previously used as Rsp5 substrate *in vitro*[50]. Our results indicate that Rsp5 could conjugate higher levels of ubiquitin moieties onto K48 at elevated temperatures, but more work would be required to further assess this possibility.

**K63 chains removal promotes proteasomal degradation after HS**. We reasoned that if the main function of the two deubiquitinases was to remove K63-linked ubiquitin on heat-induced cytosolic Rsp5 substrates, then expression of the K63R ubiquitin mutant (that cannot support the formation of K63-linkages) should bypass this requirement. We therefore assessed the degradation of [35]S pulse-labelled proteins in *UBP2* and *ubp2Δ* cells that expressed only the WT or K63R ubiquitin. Remarkably, we found that the degradation of heat-induced misfolded proteins in *ubp2Δ* cells was restored upon expression of K63R ubiquitin, with similar rates as in *UBP2* cells expressing WT ubiquitin (Fig. 7a). We confirmed that K48-linked ubiquitin could accumulate normally in this strain after HS (Supplementary Fig. 7a).

One possibility is that formation of K63-linked chains in the absence of the deubiquitinases leads to a depletion of the free ubiquitin pool, thereby preventing assembly of K48-linked chains. However, overexpression of ubiquitin itself was not

sufficient to restore the degradation of heat-induced misfolded proteins in *ubp2Δ* or *ubp3Δ* cells (Supplementary Fig. 7b). In contrast, overexpression of K48R ubiquitin (in the presence of endogenous ubiquitin) led to reduced degradation of misfolded proteins (Supplementary Fig. 7c) in WT cells. These results strongly support a model in which Ubp2 and Ubp3 further process Rsp5 cytosolic HS-induced substrates to promote conjugation of non-K63-linkages such as K48 for proteasomal degradation.

**Lysosome degradation of misfolded proteins without Ubp2 or 3**. Due to the role of Ubp2 and Ubp3 to enable proteasomal degradation of misfolded proteins by preventing the assembly of K63 chains, we reasoned that the two deubiquitinases should be dispensable for the lysosomal degradation of other quality control pathways that rely on Rsp5. We first assessed the plasma membrane surveillance pathway by evaluating the turnover of Can1 and Mup1, two membrane proteins degraded by the lysosome in an Rsp5-dependent manner upon heat-induced misfolding[27]. Endocytosis of both Can1[GFP] and Mup1[GFP] was not affected after shifting the cells to a higher temperature in *ubp2Δ* and *ubp3Δ* cells (Fig. 7b). Moreover, degradation of both

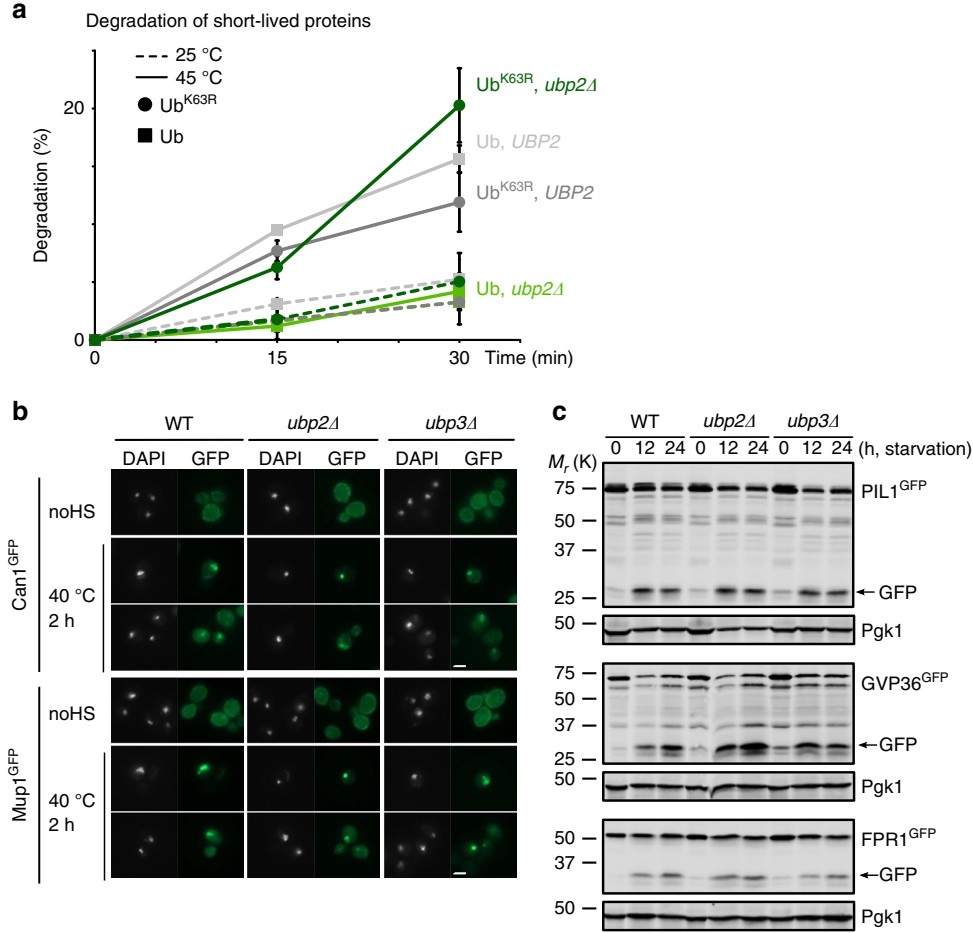

**Figure 7 | Removal of K63 chains by Ubp2 and 3 mainly promotes proteasome degradation after HS.** (**a**) Degradation of [35]S pulsed-labelled proteins in WT (grey) and *ubp2Δ* (green) cells at 25 °C (dotted lines) or 45 °C (straight lines) that expressed solely ubiquitin (light, square) or ubiquitin-K63R (dark, round). The portion of proteins degraded at the indicated times was measured and averaged for short-lived proteins in three independent experiments (with s.d.). (**b**) Representative fluorescent microscopy images of WT, *ubp2Δ* and *ubp3Δ* cells that expressed Can1[GFP] or Mup1[GFP] from their endogenous promoters and that were incubated at 25 or 40 °C for 2 h. Both the GFP and Hoechst (DNA staining) channels are shown. Scale bar, 2 μm. (**c**) Cells that expressed the indicated aggregation-prone proteins C-terminally tagged with GFP (from endogenous locus) were starved at 30 °C for the indicated times before western blots.

proteins upon HS was not inhibited in *ubp2Δ* and *ubp3Δ* cells (Supplementary Fig. 7d). We next examined the turnover of three aggregation-prone proteins that are targeted by macro-autophagy in a *RSP5*-dependent manner under starvation[21]. Deletion of either *UBP2* or *UBP3* did not appreciably affect the reduction of the assessed protein levels, as well as the accumulation of the cleaved GFP domain, which is more resistant to lysosomal proteases (Fig. 7c). These observations indicate that Ubp2 and Ubp3 are most likely solely implicated in the cytosolic Rsp5-PQC pathway, but not other Rsp5-surveillance pathways that rely on lysosomal degradation.

**Absence of Ubp2 or -3 reduces cell fitness upon heat stress.** To specifically assess the importance of the Rsp5 cytosolic PQC pathway, and the role of the two deubiquitinases, we assessed cell fitness in heat stress conditions in the absence of *UBP2* or *UBP3*. We first monitored cell growth following a 30 min HS at 45 °C,

and found that there was a prolonged recovery time in the *ubp2Δ* and *ubp3Δ* mutant cells in comparison with WT cells (Fig. 8a). We verified that the cell viability was not distinctly affected in these conditions (Supplementary Fig. 8a) to confirm that the observed lagging was due to a growth delay and not cell death. We observed similar growth delays after HS in *rsp5-1* and the double *ubp2Δ, rsp5-1* mutants cells (Fig. 8b), confirming that both the ligase and the deubiquitinase share a common—and not antagonizing—function in promoting cytosolic PQC after heat stress. As well, when assessing HS recovery in *ubp2Δ, ubp3Δ* cells, we observed a further but not additive delay compared with single mutant cells (Fig. 8c). Finally, we also observed both a reduced viability and a slower growth of *ubp2Δ* and *ubp3Δ* cells in the presence of canavanine, an arginine analogue causing protein misfolding (Fig. 8d). These data indicate that *UBP2* and *UBP3* are both required for cells to adequately respond to the increase of protein misfolding, consistent with their role in the cytosolic Rsp5-PQC pathway.

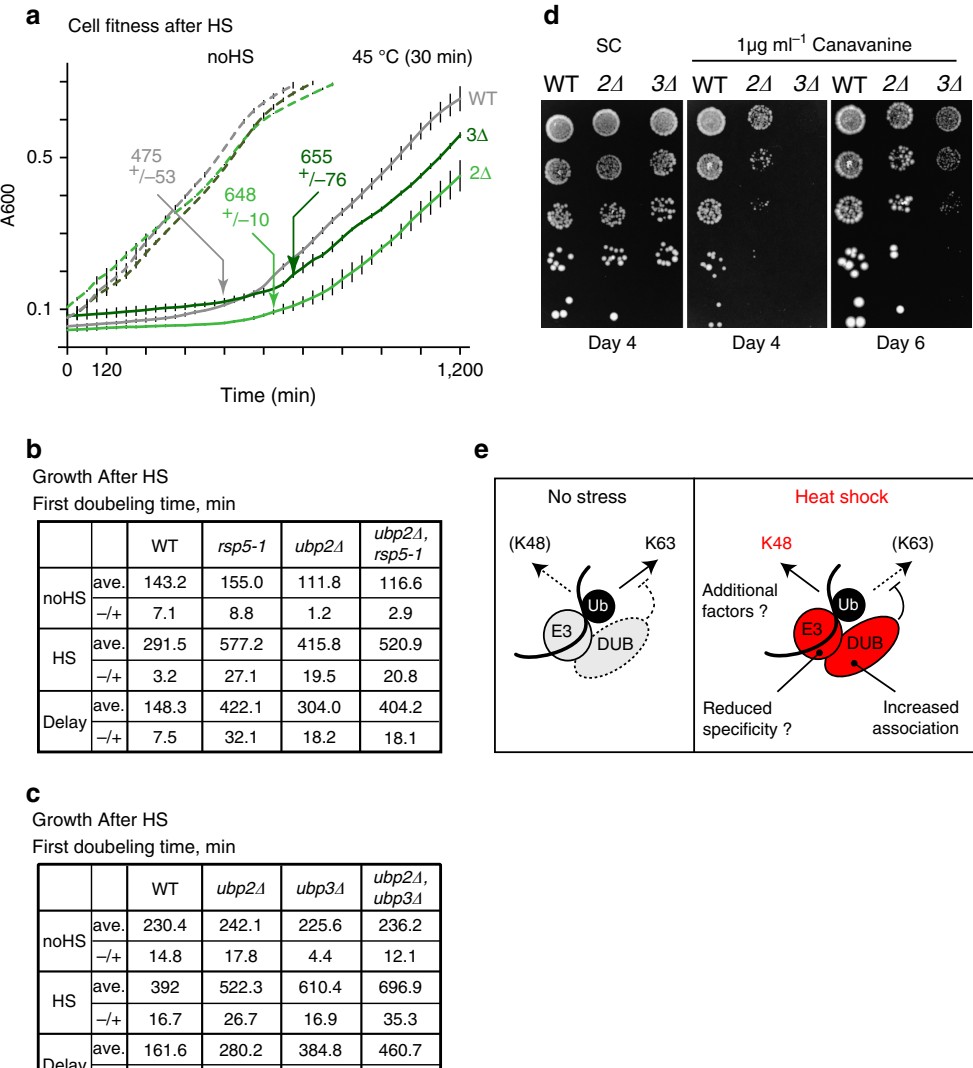

**Figure 8 | Absence of *UBP2* or *UBP3* reduces cell fitness after HS. (a)** The cell fitness of WT (grey), *ubp2Δ* (light green) and *ubp3Δ* (green) cells was measured in three biological replicates (horizontal bars represent s.d.). Cells were grown at 25 °C after no HS (dotted lines) or a HS at 45 °C for 30 min (straight lines). Time (min) required to complete the first doubling after HS is indicated for each strain with s.d. **(b,c)** Tables showing times (min) required to complete the first doubling with or without (45 °C, 30 min) treatment for the indicated cells. The experiment was done with three biological replicates ( ± s.d.). **(d)** 1/5 dilutions of WT, *ubp2Δ* and *ubp3Δ* cells grown on canavanine (1 µg ml⁻¹) and control (SC) plates at 30 °C. **(e)** Proposed model in which Rsp5 assembles K63-linked chains in unstressed conditions, while it builds up K48-linked chains upon HS to signal for proteasomal degradation.

## Discussion

Rsp5 is a major E3-ligase that targets cytosolic misfolded proteins for proteasomal degradation after heat-denaturation stress. In this study, we showed that the Ubp2 and Ubp3 deubiquitinases are also required to promote the proteasomal degradation of the misfolded proteins by preventing the assembly of K63 chains in stress conditions. Both deubiquitinases mostly reduced Rsp5-induced conjugation and displayed increased binding with Rsp5 upon HS. After prolonged HS (15–20 min), the absence of one of the two deubiquitinases did not affect site occupancy on conjugated HS-induced substrates, indicating that the two enzymes were likely involved in chain remodelling. Instead of antagonizing Rsp5 activity to reduce proteolysis of misfolded proteins, we found that the two deubiquitinases were required to remove K63-linked ubiquitin from Rsp5 heat-induced substrates to promote proteasomal degradation. We also showed that Rsp5 most likely directly conjugates the K48-linked ubiquitin on its PQC targets based on the crosslinked experiments. Interestingly, we found that Rsp5 can potentially conjugate more K48-linked chains at higher temperatures in some conditions *in vitro*. On the basis of these data, we propose a model where Ubp2 and Ubp3 play a key role in promoting the proteasomal degradation of heat-induced cytosolic misfolded proteins by limiting the assembly of non-permissive K63-linked chains, and thereby allowing the buildup of K48-linked ubiquitin by Rsp5 (Fig. 8e).

In numerous cases, deubiquitinases trim ubiquitin chains on substrates to limit their proteolysis[33,51]. Our data suggest rather that both Ubp2 and Ubp3 participate in the editing of their substrates' ubiquitin chains upon HS. The association of Ubp2 with Rsp5 has been well documented in other studies[40–42], and Ubp3 was co-purified by Rsp5 (ref. 43). *UBP2* was found to be important in promoting the endocytosis and efficient sorting of several integral membrane proteins into multivesicular bodies[40,52]. As well, deletion of *UBP2* partially rescues *rsp5-1* lethality that may be caused by osmotic instability[41]. Deletion of Ubp3 can accelerate, presumably proteasomal, degradation of RNA PolII (ref. 53) and Sec23 COPII component[44], most likely by trimming ubiquitin chains. In those cases, the two ubiquitinases functionally antagonize Rsp5. In contrast, we found that Ubp2 and Ubp3 editing activity upon heat stress positively regulated proteasomal degradation of cytosolic misfolded proteins. Therefore, the two deubiquitinases share the same function with Rsp5 in this case. Other deubiquitinases were also shown to direct the outcome of the ubiquitination reaction by chain editing[54–56], including A20 that contains both a K63-specific deubiquitinase and E3-ligase domains to promote the assembly of K48-linked chains and proteasomal degradation of RIP[57]. In our study, we uncovered a novel mechanism that relies on an increased association of the K63-specific deubiquitinases upon HS to promote proteolysis. It will be important to determine what drives the increased association of the two deubiquitinases to Rsp5 upon HS.

Intriguingly, the Hul5 ligase that also promotes degradation of misfolded proteins after HS also preferably synthesizes K63-chains[13]. 'Why is Rsp5 K63-activity detrimental to proteasomal degradation but not Hul5's activity?' Previous work from the Goldberg lab showed that proteins conjugated to K63 chains were not degraded by the proteasome in the presence of mammalian cell extracts due to ubiquitin binding proteins that preclude their interaction with the proteasome[46]. Because Hul5 is associated to the proteasome, its substrates conjugated with K63-linked chains could be processed normally. In contrast, Rsp5 substrates could be shielded by other factors preventing its recognition by the proteasome. Alternatively, the activity of the two deubiquitinases may promote Cdc48/p97 segregase activity that is required for the degradation of proteins under HS

conditions[12]. Similarly, the *C. elegans* Ataxin 3 deubiquitinase was also shown to be important in conjunction with Cdc48 for proteasomal degradation of tested substrates[58].

Upon acute HS, both Ubp2 and Ubp3 potentially act in a cooperative manner to edit Rsp5 heat-induced misfolded substrates. Deletion of either *UBP2* or *UBP3* is sufficient to fully block degradation of misfolded proteins, and to cause the association of Rsp5 to mostly K63-linked chains. As well, largely the same substrates and ubiquitination sites were affected in the absence of either deubiquitinase in the SILAC experiment. One possibility is that both deubiquitinases are simultaneously associated to the same Rsp5-containing complex to efficiently edit ubiquitin chains on a large number of Rsp5 substrates upon heat stress. Future experiments are needed to confirm this hypothesis.

Both Ubp2 and Ubp3 are involved in other stress responses. Inactivation of Ubp2 upon oxidation was recently proposed to enable the accumulation of K63 chains, including on ribosomal proteins[59]. Overexpression of Ubp3 in Hsp70 chaperones-deficient cells suppresses heat sensitivity[60], and Ubp3 was shown to mediate the clearance of JUNQ inclusion bodies in a proteasome-dependent manner[61]. It would be interesting to assess the role of Rsp5 in these pathways. Upon starvation, Ubp3 also promotes the lysosomal degradation of ribosomes via ribophagy[62]. Correspondingly Rsp5 also functions in lysosomal dependent PQC pathways[20,21,27]. Upon HS, the increased association of the two deubiquitinases likely provides a switch mechanism to repurpose Rsp5 into a proteasome targeting E3-ligase to appropriately deal with the upsurge of cytosolic misfolded proteins. Cross-talk between these PQC pathways, as well as how the deubiquitinases properly switch among their functional roles should be carefully assessed in the future.

## Methods

**Plasmid and yeast strains.** All *Saccharomyces cerevisiae* strains in S288C and SUB62 backgrounds and plasmids are listed in Supplementary Tables 1 and 2. All the single-deletion strains used in this study were from the Yeast Knockout (YKO) Collection (Open Biosystems, GE) and kindly provided by Dr Chris Loewen, with the exception of an *ubp2Δ* strain used in Figs 1b,d,f, 2a, 3a,d,e, 4a,c, 7b, and 8a-d, and *ubp2Δ* in SUB413 and SUB280 strains[63], which were generated by PCR-based homologous recombination with the HIS3MX6 cassette[64]. *UBP2-TAP* and *UBP3-TAP* strains provided by Dr Elizabeth Conibear were from the Yeast TAP-Tagged ORFs Collection (Open Biosystems, GE). All of the double-mutant strains were generated by mating and tetrad dissection. All SILAC mass spectrometry experiments were done using cells derived from the BY4742 background. *CDC19-3HA, PDC1-3HA* and *SUP45-3HA* were generated by the insertion of C-terminal 3xHA after the endogenous genes by homologous recombination[64]. *PIL1-GFP, GVP36-GFP* and *FPR1-GFP* strains are from the Yeast-GFP Clone Collection (Invitrogen) and kindly provided by Dr Phil Hieter.

pRS315-*UBP2* (BPM605) and pRS315-*UBP3* (BPM607) were generated by inserting PCR amplified *UBP2* or *UBP3* from genomic DNA (open reading frame with 500 bp up- and downstream sequences) into pRS315 plasmid using XmaI and SpeI restriction sites. The catalytic-dead mutants *ubp2-C745A* (BPM606) and *ubp3-C469A* (BPM608) were obtained by site-directed mutagenesis. *3HA-RSP5* (BPM486) was kindly provided by Dr Teresa Zodalek (Polish Academy of Sciences, Poland) and used to generate *3HA-rsp5-I537D* (BPM762) and *3HA-rsp5-C777A* (BPM835) by site-directed mutagenesis. The megaprimer method to generate BPM762 was adapted from a previously described method[65]. Briefly, a reverse primer containing the mutated sequence was used to PCR amplify the upstream region of the mutation site within the *RSP5* sequence of BPM486 to generate megaprimers, which were then used to PCR amplify BPM486 before DpnI digestion and bacterial transformation. The *CDC19-13MYC* (BPM563), *cdc19-D367R-13MYC* (BPM 625), His$_8$-ubiquitin (BPM30), single MYC-tagged ubiquitin expression constructs K48-only (BPM591) and K63-only (BPM592) were generated as previously described[14]. *GST-UBC4* (BPM746), *GST-UBP2* (BPM740) and *GST-RUP1* (BPM741) were kindly provided by Dr Jon Huibregtse (University of Texas at Austin, USA); GST-TRIM5α(RING1-93), BPM910, was a kind gift from Dr Dimitri Ivanov (University of Texas Health Science Center). The *GST-RSP5* (BPM98) was generated as previously described[66]. *GST-CDC19* (BPM581) was generated by subcloning *CDC19* from pRS313-*CDC19-13MYC* (BPM563) into pGEX-6P-2 using BamHI and SalI restriction sites. The pRS416-*CAN1-GFP* and pRS416-*MUP1-GFP* (BPM 552 and 553) were kindly provided by Dr Elizabeth Conibear. The *13MYC-MUP1* (BPM596) and *13MYC-CAN1* (BPM595) plasmids

were generated by PCR amplification of the ORFs from *MUP1* and *CAN1* containing plasmids (BPM 552 and 553) and inserted into BPM173 (pRS313 with *GPD* promoter, 13xMYC tag and *PGK* terminator sequence) using the NotI site. pRS316-5′*UTR-3HA-HUL5*-3′*UTR* (BPM812) was generated using the MEGAWHOP method as previously described[65]. The 3HA sequence from pFA6a-3xHA-*His3MX6* was PCR amplified to generate the megaprimer, which was then used to amplify pRS316-5′*UTR-Hul5*-3′*UTR* (BPM752) to add an N- terminal 3HA tag.

Mouse monoclonal anti-ubiquitin antibody MAB1510 (1:3,000; Millipore) was used for assessing ubiquitin levels by western blots and quantitative dot-blot assays, as well as the anti-K63 (1:1,000; #05-1308 from Millipore) and anti-K48 (1:1,000; #8081 from Cell Signaling). The house-keeping protein Pgk1 (3-phosphoglycerate kinase) was used as a loading control and was detected using a rabbit polyclonal anti-Pgk1 antibody (AP21371AF-N, 1:10,000) from Acris Antibodies. Mouse monoclonal anti-MYC (9E10, 1:5,000), and anti-HA (12CA5, 1:2,000) antibodies from the AbLab UBC in-house facility were used to detect MYC- and HA- tagged proteins. Polyclonal anti-TAP (CAB1001, 1:1,000) from Pierce (Thermo Scientific) and anti-GFP (11 814 460 001; 1:1,000) from Roche were used to detect TAP- and GFP-tagged proteins. The goat anti-Rsp5 antibody (sc-26193, 1:1,000) from Santa Cruz was used to detect endogenous Rsp5. Anti-mouse-800, anti-rabbit-700 and anti-goat-800 fluorescent secondary antibodies (1:10,000; LI-COR) were detected with an Odyssey Infrared Imaging System (LICOR). Protease inhibitor cocktail was purchased from Roche, and all other reagents were purchased from Sigma unless specified (except reagents for yeast media; Thermo Scientific).

**HS and protein stability assays.** After HS, cells were collected and processed as previously described[17]. For measuring ubiquitination levels in HS experiments, overnight-saturated cultures were diluted and grown to exponential phase in YPD or synthetic media at 25 °C to an $A_{600}$ of 1–1.5, ~2 ml of cells were then incubated at 45 °C (if not indicated otherwise) in a thermomixer for the indicated times (5–15 min). For western and dot-blots, cells were snap frozen in liquid nitrogen and washed twice with ice-cold 1 × TBS (50 mM Tris 150 mM, NaCl pH 7.5) before lysis. Cells were then resuspended in pre-warmed 1 × SDS–polyacrylamide gel electrophoresis (SDS–PAGE) Laemmli sample buffer without reducing agent and dye and incubated at 94 °C for 2 min before the lysis with glass beads in the Precellys 24 tissue homogenizer (Precellys). All samples were normalized after using a Bradford assay (BioRad). For dot-blot assay, 3 μl of the equalized samples (5–10 μg proteins) was spotted and dried overnight on nitrocellulose membranes, which were rehydrated with 1 × TBS for 10 min and processed as other western blots. All quantitative dot-blots were performed by measuring signals from three biological replicates except for the initial deubiquitinases HS screening. For each sample, the ubiquitin signal was normalized using the Pgk1 signal in a multiplexed analysis using the Odyssey system (LI-COR). When reporting normalized ubiquitination levels, the dot-blot signal intensities were averaged across the three samples and normalized to the averaged ubiquitination level in the reference sample (typically WT) at 25 °C, which was set to the arbitrary value of 1. When reporting the increase in ubiquitination, the difference of signal intensities (normalized to Pgk1 levels) between the two temperatures for each sample (for example, 45 °C–25 °C) was calculated, then averaged across the three replicates, and normalized to the averaged difference in the reference sample (set to the arbitrary value of 1). For quantifying poly-ubiquitination signal from western blots, regions above 75 kDa were typically used.

***In vivo* crosslinking and co-immunoprecipitation experiments.** For all IP experiments in Figs 2c,d and 5b-f, *in vivo* crosslinking was done with 1% formaldehyde. All IP experiments in the Supplementary Figures were also done with crosslinked cells except the Cdc19-13MYC pulldown experiment in Supplementary Fig. 6E. Cells were grown at 25 °C to an $A_{600}$ of 1 in synthetic media and then heat shocked at 40 °C for 10 min before the addition of 1% formaldehyde for an additional 10 minutes (unstressed cells were maintained at 25 °C for the crosslinking). Cells were then placed on ice, and the crosslinking reaction was quenched with an excess of glycine (~250 mM) for five minutes at 4 °C. The samples were centrifuged, washed twice with cold 1 × TBS, and then frozen in liquid nitrogen. For the Cdc19-13MYC IP, *Tetp::RSP5* cells were kept in synthetic media with 100 μg ml⁻¹ doxycycline and 0.1% TWEEN 80 before the experiment and cells in logarithmic phase were heat shocked at 45 °C for 20 min before harvest. Cells were lysed with modified RIPA buffer (50 mM Tris-HCl pH 7.5, 150 mM NaCl, 1% NP40, 0.5% sodium deoxycholate, 0.5% SDS, 1 mM PMSF, 10 mM chloroacetamide, 1 mM Phenanthroline, and 1 × protease inhibitors cocktail) using glass beads. Lysates were then slowly diluted to 0.1% SDS final for IPs. Pellets were resolubilized in lysis buffer containing 0.1% SDS at room temperature for 30 min to maximize the proteins recovery. Sample concentrations were measured by Bradford assay and equalized before the overnight incubation with beads at 4 °C. Beads were washed six times in lysis buffer with 300 mM NaCl and eluted with 1 × SDS buffer without reducing agent to avoid elution of crosslinked IgG. Reducing agent was added and samples boiled at 94 °C for 30 min to reverse crosslinking before running on a SDS–PAGE gel. All Western blots were analysed with the Odyssey system (LI-COR). Uncropped images of main Western blots are shown in Supplementary Fig. 9. Nickel, anti-HA, and anti-MYC magnetic beads

were purchased from Promega, Pierce (Thermo Scientific), and Origene respectively.

**Ni²⁺ purification of ubiquitinated proteins.** For immobilized metal ion affinity chromatography (IMAC), about 1 μl of MagneHis (Promega) was used per 200 μg of protein extract. A total of 150 ml of cells carrying H₈-Ubi plasmid (BPM30)[67] or corresponding empty plasmid for control were grown in SC-URA media at 25 °C. Cells with or without 20 min HS treatment at 45 °C were washed twice with cold 1 × TBS and snap frozen in liquid nitrogen. Thawed cells were lysed in HU buffer (8M urea, 100 mM HEPES at pH 8, 0.05% SDS, 10 mM chloroacetamide, 1 mM PMSF, 10 mM imidazole and protease inhibitors cocktail) by glass beads. Following 90 min incubation with cell extracts at ambient temperature, nickel beads were washed three times in HU buffer with 1% SDS. Bound proteins were eluted by incubating the beads in one volume of 8M HU and one volume of 2M Imidazole for 10 min at ambient temperature with shaking. 1/3 volume of 3 × SDS–PAGE Laemmli sample buffer was added before heating elution at 70 °C for western blot analysis or in-gel digestion.

**³⁵S-labelling and protein turnover assays.** Quantification of ³⁵S-labelled protein degradation was performed as previously described[12,17]. Yeast cells grown in SC medium (SC denotes a synthetic complete media with all amino acids and 2% dextrose while the SD medium is a minimum media with only ADE, URA, TRP, HIS, MET, LEU, LYS and 2% dextrose) with additional tyrosine (20 g l⁻¹) to an $A_{600}$ of 0.8–1 were washed and incubated in SC-MET media with additional tyrosine (20 g l⁻¹) for 50 min. EXPRE35S35S protein labelling Met & Cys mix (50 μCi per m1, PerkinElmer) was added for 5 min incubation before washing cells with ice-cold SC-chase media containing cysteine (0.5 mg ml⁻¹), methionine (6 mg ml⁻¹) and cycloheximide (0.5 mg ml⁻¹). Cells were then incubated at indicated temperatures. Equal volume of culture was collected at indicated times and mixed to a final concentration of 10% TCA. After an overnight incubation at 4 °C, radioactivity in both TCA-soluble and -insoluble fractions was measured in a MicroBeta2 radiometric detector (PerkinElmer). The percentage of protein degradation was calculated by subtracting the signal from TCA-soluble at time 0 from indicated time that was then divided by the signal in the TCA-insoluble fraction at time 0. All the experiments were all repeated three times and the average protein degradation rates are reported. For the experiment with overexpression of His₈-ubiquitin under the *GPD1* promoter, cells with the BPM30 plasmid were grown in SD-URA then SD-MET-URA with additional tyrosine before labelling. Turnover of the MYC-tagged Cdc19-D367R was assessed in SC-HIS media. Cells were grown at 25 °C before the addition of 100 μg ml⁻¹ cycloheximide and incubation at 42 °C. Turnover of MYC-tagged Can1 and Mup1 plasma membrane proteins was assessed in SC-HIS media. Cells were grown at 25 °C before the addition of 100 μg ml⁻¹ cycloheximide and incubation at 40 °C. Turnover of GFP-tagged aggregation-prone proteins was assessed after shifting cells grown in YPD media to log phase then switched to SD media without ammonium sulfate and with 2% dextrose after two washes in the same SD media at 30 °C.

**DiGly peptide enrichment for triple-SILAC analysis.** Enrichment of diGly containing peptides was performed using the PTMScan kit (Cell Signaling) with minor modifications from previous descriptions[38]. Only lysine (K) labelling was used for the diGly SILAC experiments. Cells were grown in small volume of SC-LYS media supplemented with the indicated isotopically labelled lysines (0.3 mg l⁻¹) till saturated at 25 °C and diluted in large volume of the same media to an $A_{600}$ of 0.2. Cells were grown to an $A_{600}$ of 1. For the experiments in Fig. 4b,c, *ubp2Δ* or *ubp3Δ* cells labelled with light-lysine (K0) and WT cells labelled with median-lysine (K4; D₄) were subjected to 20 min HS at 45 °C before lysis, while heavy-labelled WT cells (K8; D₂,¹³C₆¹⁵N₂) were kept at 25 °C for control. For the experiment in Supplementary Fig. 4B, *ubp2Δ*, WT and *ubp3Δ* cells were labelled with K0, K4 and K8, respectively, and heat shocked at 45 °C for 20 min before harvest. Longer HS incubation periods were required (20 min versus 15 min), as larger volumes of cells were used in comparison with other experiments. Cells were lysed in 8 M urea lysis buffer (8 M urea, 50 mM Tris-HCl and 150 mM NaCl at pH 7.5, 10 mM chloroacetamide, 1 × protease inhibitors cocktail, 1 mM sodium orthovandate, 2.5 mM sodium pyrophosphate and 1 mM β-glycerophosphate). Equal amounts of lysate from differentially labelled cells were mix together, to obtain ~30 mg proteins in total. The lysate was reduced in 3 mM TCEP (3,3′,3″-phosphanetriyltripropanoic acid) at 25 °C for 30 min and then was alkylated in 55 mM chloroacetamide at 25 °C for 45 min in dark. 50 mM Tris-HCl pH 7.6 was then used to slowly dilute lysate to final concentration of urea under 2 M with addition of CaCl₂ to final 1 mM before trypsin digestion. 300 μg of trypsin (1/100) was used to digest the lysate at 30 °C for 36 h. Digestion was stopped by adding 1.5% formic acid and incubated at 25 °C for 10 min before 10 min centrifugation at 15 °C, 15,000g. Acetified peptides were cleaned on two high capacity C18 columns (Thermo). Peptides were eluted from each column using three times 2 ml 50% acetonitrile elution buffer. The PTMScan Ubiquitin Remnant Motif Kit (Cell Signaling Technology) was used to immuno-precipitate (IP) diGly peptides. Speed-vac dried peptides were resuspended in 1.4 ml PTMScan IAP buffer (50 mM MOPS · NaOH pH7.2, 10 mM Na₂HPO₄, 50 mM NaCl) for 2 h at 4 °C after brief sonication. IP was performed according to the manufacturer

protocol, except 1/8 of the bead for one IP was used and beads were crosslinked before IP[68]. Briefly, antibody-bound beads were washed three times with wash buffer (100 mM sodium borate, pH 9.0) before cross-link antibodies to the beads for 30 min at 25 °C in crosslinking buffer (20 mM DMP in 100 mM sodium borate, pH 9.0). Crosslinking was stopped by washing twice with antibody-blocking buffer (200 mM ethanolamine pH 8.0) and 2 h incubation in blocking buffer at 4 °C. before adding lysate, the beads were washed three times in IAP buffer. IP was done at 4 °C for 1 h. Beads were washed twice with IAP buffer and three times with PBS by gently inverting tubes five times. Bound peptides were eluted with three times of 50 µl 0.15% (vol/vol) Trifluoroacetic acid at 25 °C for 5 min before being cleaned up by C18 stage tips without fractionation[69].

**Tryptic digest for ubiquitin linkage quantification by SRM.** For SRM analysis, IMAC of His$_8$-ubiquitin and HA-IP eluted samples were subjected to in-gel digestion. Gel bands, which were above 75 kDa (poly-ubiquitination fraction) from each IMAC sample and were above the molecular weight of untagged substrate from each HA-IP sample, were cut out for further processing and trypsin digestion. Briefly, gel was fixed and stained in coomassie staining solution for 30 min before washing in ddH$_2$O for 3 h. Gel bands were then excited and cut into small cubes for distaining three times in 50 mM NH$_4$HCO$_3$/ethanol (1:1) for 20 min. Ethanol-dehydrated gel pieces were incubated with 10 mM DTT in ddH$_2$O at 56 °C for 45 min following by 30 min in 55 mM chloroacetamide (in 50 mM NH$_4$HCO$_3$) at ambient temperature. Gel pieces were washed three times in 50 mM NH$_4$HCO$_3$ for 20 min with dehydration steps between washes. Dehydrated samples were then incubated with trypsin solution (12.5 ng µl$^{-1}$ in 50 mM NH$_4$HCO$_3$) on ice for 60 min before starting the digest at 37 °C for 18–22 h. Digested peptides were extracted by extraction buffer (0.5% acetic acid) with increasing amount of acetonitrile till the gel pieces were completely dehydrated. Combined extracted liquid from each sample was dried and resuspended in Buffer A (0.5% acetic acid) for purification with C18 stage tips without fractionation. Equally amount of ubiquitin AQUA peptides (2.5 µl of 10 pmol µl$^{-1}$ 48GG, 63GG) were digested in 100 mM Tris-HCl pH 8.5 with 0.2 µl of 0.2 µg µl$^{-1}$ trypsin at 37 °C overnight. Following C18 stage tip purification, equal volume of AQUA peptides was spiked-in all in-gel digested samples for analysis on a QQQ instrument.

**Mass spectrometry analyses.** For diGly sites mass spectrometry analysis, purified peptides were analysed using a LTQ-Obitrap Velos (ThermoFisher Scientific) online coupled to an Agilent 1,100 Series nanoflow high-performance liquid chromatography (HPLC) using a nanospray ionization source (ThermoFisher Scientific). Samples were run with a 120 or 180 min gradient with a full-range scan at 60,000 resolution from 350 to 1,600 Th and HCD ionization method to fragment the top five peptide ions in each cycle in the LTQ (minimum intensity 2,000 counts). Parent ions were then excluded tandem mass spectrometry for the next 30 s as well as singly charged ions. The Velos was continuously recalibrated using the lock-mass function[70]. Raw files were subjected to MaxQuant (version 1.5.1.0) for peptides ID searching against the *Saccharomyces* Genome Database (SGD-05 Jan 2012 with 6,147 protein sequences and 6,147 randomized sequences) and peptide quantification. For each data set, peptide ratios were normalized to the median protein ratios obtained from corresponding whole-cell lysate aliquots. Similar to previous studies, we removed peptides with a C-terminal diGly and considered peptides with log$_2$ rations ≥ 1 (refs 45,71). Cellular localization was based on a previous analysis[72].

For SRM analysis, peptides (K48GG, K63GG) were purchased from JPT Peptide Technology and each peptide was reconstituted in 0.1% acetic acid to yield 10 pmol µl$^{-1}$ for storage at −80 °C. C-terminal JPT-tag can be removed by trypsin digest and peptides are labelled with terminal heavy residue (Lysine: +8; Arginine: +10). SRM analysis was done using a triple quadrupole mass spectrometer G6460 (Agilent Technologies) online coupled to an Agilent 1,200 Series HPLC using an Agilent ChipCube nanospray ionization source. A large capacity protein chip (Agilent Technologies) 150 mm 300A C18 column with 160 nl trap column (both with C18) was used. A 34 min gradient method adapted from a previous study was used[73]. The samples were loaded with an Agilent 1,200 Cap Pump at 4 µl min$^{-1}$ and the analytes were eluted with a 1,200 nano-pump at 300 nl min$^{-1}$. The most abundant charge states of heavy and light versions of each peptide were determined empirically and used for SRM transition development as previously described and SRM transitions (fragmented ions) with the top three highest intensities were selected to monitor the parental ion[73]. Peptides were monitored throughout the whole HPLC gradient. For data generated, peak integration was performed using MassHunter Quantitative Analysis software (Agilent Technologies, version B04.00, build 4.0.479.0), and area under the curve was used to determine the abundance of each light peptide in the sample relative to its corresponding heavy internal standard[73].

**Microscopy.** Cells were grown to an A$_{600}$ of 1 at 25 °C and subjected or not to a 40 °C HS for 2 h. 2.5 µg ml$^{-1}$ of Hoechst 33342 (Invitrogen) was added to all cells in the last 30 min of incubation to stain the nuclei. Live cells were washed twice in 1× PBS and mounted on a slide with 1× PBS and immediately imaged with an inverted Zeiss Axio Observer microscope with a ×63 oil objective and processed with Zeiss Zen software.

**Recombinant protein purifications.** Overnight-saturated BL21(DE3) cells carrying GST-tagged gene containing plasmids (pGEX6p-1, pGEX6p-2 or pGEX-4T-1) were diluted 100 times in LB media and incubated at 30 °C until A$_{600}$ reached 0.8. For expression of GST-Rup1, GST-Ubc4 and GST-Trim5α (RING1-93), 0.8 mM IPTG was added for induction for 5 h at 25 °C. Induction of GST-Rsp5 and Ubp2 were carried out at 16 °C for 22 h with 0.8 mM IPTG. Cells were collected and washed twice with cold 1× TBS and snap froze before lysis. Cells were incubated with 5 µg ml$^{-1}$ DNAse and 5 mM MgCl$_2$ in native lysis buffer (1× PBS pH 7.3, 10% glycerol, 1 mM DTT, 1 mM PMSF, 1 mM EDTA, 1× Roche Protease Inhibitor Cocktail) before lysing by sonication. Cells containing GST-Trim5α(RING1-93) were incubated with 2.5 mg ml$^{-1}$ lysozyme in lysis buffer (Tris-HCl pH 8.0, 10% glycerol, 1 mM DTT, 1 mM PMSF, 100 mM NaCl, 1× Roche Protease Inhibitor Cocktail) before lysing by sonication. 1% Triton X-100 was added to sonicated lysates before centrifugation at 15,000g for 30 min. Cell lysates were then loaded onto a gravity column with Glutathione Sepharose 4B beads (GE) and incubated for 4 h at 4 °C. Bound tagged proteins were washed 6 times with 10× bed volume of lysis buffer before GST tag cleavage on beads using PreScission Protease (GE) for 4 h at 4 °C or thrombin (for GST-Trim5α) for 6 h at room temperature. Concentrations of cleaved proteins were determined using Nano-drop 1,000 spectrophotometer (Thermo Scientific).

Short WBP2-1K (residues 169–256) with an extra cysteine at N-terminus that contains single lysine residue (K222) was generated. The construct was cloned into pRSF-duet vector fused with MBP-TEV-His tag at the N-terminal end. The fusion protein was overexpressed in *Escherichia coli* RIL strain. Nickel affinity pulldown, anionic exchange, TEV protease digestion and size exclusion chromatography were applied sequentially to obtain pure His-tagged short WBP2-1K that was then concentrated, flash-frozen by liquid nitrogen and stored at −80 °C. Crosslinking reaction of fluorescein (AnaSpec, Fremont CA) and WBP2 was carried out as previously described[74].

**In vitro ubiquitination assay.** GST-Ubc4, GST-Rsp5, GST-Ubp2, GST-Rup1 and GST-Trim5 were purified from bacteria followed by GST cleavage. The E1 (Ube1) and the Myc-tagged or His$_6$-tagged (WT or mutant variants) ubiquitin were purchased from Boston Biochem. In vitro ubiquitination assays were carried out as previously described[28] with slight modifications. The reaction buffer consisted of 25 mM Tris-HCl pH 7.6, 50 mM NaCl, 5 mM MgCl$_2$, 0.5 mM DTT and 5 mM ATP. His$_6$-tagged or MYC-tagged WT and His$_6$-tagged mutants of ubiquitin, E1 (Ube1), as well as UBE2N/UBE2V2 were purchased from Boston Biochem. Reactions were carried out in 20 µl volume with 1 µg ubiquitin, 10 nM of E1, 150 nM of Ubc4, 100 nM of Rsp5, Ubp2 and Rup1. The Trim5α reactions were carried out in 20 µl with 1 µg ubiquitin, 10 nM of E1, 150 nM of UBE2N/UBE2V2, 100 nM of Trim5α. In vitro Wbp2 ubiquitinations were carried out in 30 µl on a thermocycler at 30 or 39 C. Reactions were initiated by mixing 30 µM Ub (K48 only, or K63only) with premixed 50 nM E1, 500 nM Ubc4, 100 nM Rsp5 and 300 nM fluorescein-conjugated Wbp2, while both fractions were incubated at specified temperature 5 min prior mixing. Phosphoimager Typhoon 9500 (GE Health Science) was used to scan the fluorescent signals. All samples were prepared on ice with ubiquitin added last before the incubation at different temperatures for indicated time. All reactions were stopped by snap freezing samples in liquid nitrogen followed by the addition of 3× SDS–PAGE sample buffer. SDS-containing samples were then boiled before western blot analysis.

**Yeast fitness and viability assay.** For plate assays, exponentially growing cells were diluted to a concentration of 0.5 × 10$^6$ cells per ml and 3 µl were plated in a 1/5 dilution series on YPD or SC plates. For culture fitness assay, exponentially growing cells in YPD media were diluted to an A$_{600}$ of ∼0.2 and incubated for 30 min at 25 °C or 45 °C and then transferred to a 96-well plate placed in an Infinite 200 PRO plate reader system (Tecan) with constant shaking at 25 °C. Three biological replicates were analysed in the same experiments. For each well, nine measurements (3 × 3) were taken every 30 min and averaged.

**Data availability.** The mass spectrometry data have been deposited in the PRIDE archive and are available via ProteomeXchange with the identifier PX004747. The data that support the findings of this study are also available from the corresponding author upon request.

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

## Acknowledgements

We are grateful to all colleagues cited in the method section who provided reagents, to Brenda Schulman (St Jude) for discussions and support, Ray Deshaies (Caltech) for suggestions, and to Thorsten Hoppe (U. of Cologne) and Daniela Rotin (SickKids) for their comments on the manuscript. We thank Dr N. Stoynov (UBC) for his help with mass spectrometry experiments and members of the Mayor lab for discussions. This work was supported by a grant from the Canada Institutes of Health Research (CIHR) and New Investigator Career Awards from CIHR and Michael Smith Foundation for Health Research.

## Author contributions

N.N.F. designed most of the experiments through discussions with T.M. N.N.F. carried out most of the experiments. M.Z. prepared several plasmids and strains, helped with and performed some of the experiments. A.R. prepared several reagents, helped with and performed some of the experiments. K.-P.W. performed one experiment with Wbp2. N.N.F. and T.M. wrote the paper and all other authors edited or commented on the manuscript.
