## [Peer Review File · Nature Communications]

Reviewers' Comments:

Reviewer #1 (Remarks to the Author)

The manuscript by Fang et al titled "Degradation of misfolded cytosolic proteins by the proteasome upon heat-stress relies on editing activity of deubiquitinases" explores how the E3 ubiquitin ligase Rsp5/NEDD4 can deliver substrates to the proteasome for heat-induced degradation but also to the vacuole/lysosome for regulated degradation. Importantly, the authors find that two deubiquitinating enzymes (DUBs) in yeast - Ubp2 and Ubp3 - interact with Rsp5 and provide a chain-editing function that switches the linkages of ubiquitin on the substrates from K63 linked to favor K48 linked during heat shock (HS). This allows for proteasomal degradation of misfolded proteins during HS.

Overall, I find the experiments well conducted, the statistics sound where applied, and the conclusions are original. I think the manuscript has the potential to influence thinking in the field because E3s and DUBs are often thought to work in opposition to each other, not in coordination. This is a strong paper showing the cooperativity of two opposing enzymatic actions. I support publication of this manuscript should the authors address the specific points below.

Specific points:

1. In Figure 1A, the authors saw considerable changes in ubiquitination levels after deletion of UBP2 and UBP3. A few other DUB deletions looked promising as well, like *ubp7Δ* and *ubp10Δ*. Did the authors test these other DUB delete strains in westerns like supplemental figure 1c-d? It might be worth having those as supplemental too, though lack of the data should NOT distract from the overall conclusions of the manuscript. This is just an interested desire by this reviewer to see the data and is NOT essential for publication.
2. I don't understand the labeling in Figure 1d-f. Did the authors do the experiment where they go from 45C to 25C for 5 min? Or is it from 25C to 45C for 5 min? I think it is probably the latter, but the authors should clarify here. This applies to Figure 2a-c as well. I think the authors need a better description of the experiment in the figure legends and results text to help in their shorthand in the actual figure. Maybe they could explicitly state this is the ratio of ubiquitin conjugates at 45C versus 25C normalized to wt?
3. The authors should note if they performed a step to reverse the formaldehyde crosslinking in Figure 2d-e prior to the gel analysis but after the co-purification. The levels of the Ubp2- and Ubp3-TAP pulldowns are lower after crosslinking, suggesting that they did not because formaldehyde would most likely interfere in the binding of the TAP tag to the column. The results are convincing in the co-purification of Rsp5, but I don't think the general reader appreciates formaldehyde crosslinking and the effect that the lack of reversing it can have on purifications.
4. In Figure 3, why did the authors switch to 38C as the HS rather than 45C? It would be good to explain this shift in methodology. It wasn't obvious why they did.
5. In Figure 4C, was the His8-ubiquitin construct overexpressed? It would be important to note that in the results section and the figure legend. It doesn't pose a problem for the interpretation of the results, but the general reader might like to know.

6. On page 9, the authors state "We co-expressed HARsp5 and Myc-tagged ubiquitin mutants containing only K48 or K63 in cells that were or not subjected to HS." I think it would be good to clarify that all other K residues were mutated to R for this experiment. That would not be clear to a general reader.

7. I thought the in vitro experiments were great! I only wish Figure 6a had the time course in order with each K linkage. Though, others would disagree with me. The data is solid.

8. On page 12, the authors have a section titled "Absence of Ubp2 or Ubp3 reduces cell fitness upon stress". I would caution them to be careful and just say heat stress here. Or, they could say 'cytoplasmic misfolding stress'. I know they show canavanine stress (which is probably misfolding off the ribosome and cytoplasmic), but what about ethanol, osmotic, reducing, oxidative stresses to name a few? There are lots of other stresses where the phenomenon they describe might not have an influence. I think caution to not overstep their data in a conclusion is a good thing here.

9. While I find the manuscript generally easy to follow, there are a few misspellings and poor verb tense uses scattered throughout. It would be good to have a fresh eye edit the manuscript once it is revised.

Reviewer #2 (Remarks to the Author)

Fang et al. report on the requirement of two ubiquitin hydrolases (DUBs), Ubp2 and Ubp3, for proteasome-mediated degradation of proteins in heat-shock (HS)-exposed yeast cells. The authors report that HS-treated cells lacking either DUB display a significant accumulation of poly-ubiquitylated material. At the same time, a reduced rate of degradation of short-lived proteins is observed. When cells are exposed to HS, increased amounts of epitope-tagged variants of both DUBs can be cross-linked to Rsp5. The Rsp5 ubiquitin (Ub) ligase is linked to the increased poly-ubiquitylation of proteins after HS. In unstressed cells ubiquitin chains linked via lysine 63 (polyUB-K63) can be crosslinked to Rsp5, whereas ubiquitin linked via lysine 48 (polyUB-K48) is found at the ligase after HS. Intriguingly, heat-stressed cells deleted for UBP2 or UBP3 do not accumulate polyUB-K48 but rather polyUB-K63 at the ligase. Finally, the authors show in vitro that Rsp5 preferentially synthesizes polyUB-K63 at lower temperatures but gains the ability to form polyUB-K48 at higher temperatures. As determined in genetic assays both DUBs protect cells from the deleterious effects that evolve from the accumulation of misfolded proteins. From these data the authors conclude that at elevated temperatures Rsp5 gains the ability to generate polyUB-K48. The DUBs Ubp2 and Ubp3 further support polyUB-K48 production by disassembling Rsp5-generated ubiquitin-K63 linkages and thereby facilitate the proteasomal degradation of heat-denatured proteins.

All experiments are of more than adequate technical quality and contain appropriate controls. The authors observe a highly interesting modulating role of DUBs that in combination with possible changes in the catalytic capabilities of a ubiquitin ligase causes a major re-arrangement of the ubiquitin landscape within cells in response to external stress. These findings should be of utmost interest to a broader readership. However, I have some concerns about the significance of some of the data. The nature of the physical Rsp5-Ubp2/Ubp3 interaction is unclear up to this point (see below). This point is critical for the understanding of the function of both DUBs in HS protein quality control. Furthermore, it is mandatory that the authors conduct additional experiments to investigate the putative temperature-induced switch in the Rsp5 activity in more detail. Should the authors succeed to address these points appropriately, I recommend publication of this interesting study in "Nature Communications".

Major issues.

Fig. 2d,e: There is significantly less Ubp2-TAP and Ubp3-TAP detected in cell lysates after a relatively short exposure to heat for 20 min., whereas the amount of HA-Rsp5 and Pgc1 appear to be merely unchanged. This indicates that the TAP-tagged proteins are rapidly degraded upon HS and, as a consequence, that Ubp2-TAP and Ubp3-TAP may represent substrates of Rsp5 rather than functional interaction partners in HS-exposed cells. Since an increased association of the DUBs with Rsp5 after HS is one of the proposed mechanisms to explain the function of these enzymes in promoting polyUB-K48 synthesis, it is critical to carefully address this issue. The authors should determine Ubp2-TAP and Ubp3-TAP protein levels in cells that are impaired in Rsp5 or proteasomal function. They should also monitor the amount of endogenous Ubp2 and Ubp3 that can be purified with Rsp5 under various conditions to rule out any impact of the TAP tag.

Fig. 6: Additional experiments are required to allow a proper interpretation of the results shown in this figure. Fig. 6a: There are several issues to consider here. The overall synthesis of poly-ubiquitin chains increases with raising temperature. Moreover, it is unclear, whether the K48 and K63 K-only variants of ubiquitin display distinct intrinsic propensities for the incorporation into Ub chains at the employed temperatures. Thus, control ubiquitylation reactions containing an unrelated ligase (e.g. Hul5) are required to assess, which of the temperature-dependent changes in ubiquitin chain formation can be attributed to Rsp5.

Fig 6b: The specificity of the used antibodies for a certain type of ubiquitin chain is critical for the interpretation of this assay. Hence, a control experiment demonstrating the reactivity of these antisera towards defined ubiquitin chains should be included. Again, monitoring the activity of an unrelated ligase, like Hul5, could help to ascertain that the observed effects can be assigned to Rsp5.

Overall, a more reliable quantification of K63 and K48-linked poly-ubiquitin formation by Rsp5 at different temperatures should be included, e.g. by selected reaction monitoring (SRM) as shown in Fig. 4. Again, ubiquitylation reactions by an unrelated ubiquitin ligase must be included as controls.

Reviewer #3 (Remarks to the Author)

Summary:

The manuscript by Mayor and colleagues focused on the protein quality control pathway in yeast. The authors have extended their analysis for the role of the yeast Nedd4 ubiquitin ligase Rsp5 targeting misfolded cytosolic proteins for proteasomal degradation during heat-shock. In the present study, the authors identified two deubiquitinases Ubp2 and Ubp3 functioning in the process. They provide evidence that these two DUBs prevent the formation of K63-polyubiquitination on Rsp5 substrates and thus promote their proteasomal degradation. While this work was rather detailed and comprehensive for both in vivo and in vitro analyses, the interpretations for some data and the conclusion about the "editing activity of DUBs" require additional discussion and clarification. Alternative models and/or more supporting evidence are needed prior to publication in Nature communications.

Major concern:

1. The authors need to address why in Figure 1-3, they chose to use short (5min) heat-stress, but in Figure 4 and 5 they used a longer heat-stress condition (15-20min). The difference of the site occupancy in Figure 3e and S3 also requires more clarification.

a. Based on Fig S3, the authors argue that *ubp2Δ* or *ubp3Δ* does not lead to ubiquitination of different pools of substrates. It also seems to me that these results further suggest that for a given substrate, no additional lysine residue is modified in *ubp2Δ* or *ubp3Δ* and therefore the increased ubiquitination levels are likely due to the attachment of longer ubiquitin chains "on the same residues". I

presume this is the reason the authors put "mixed K48 and K63" conjugations in the Figure 5a editing model. If this is indeed what the authors wanted to suggest, they need to clarify this in the text.

b. However, the authors also show in Figure 3e that during short (5-min) heat stress, the diGly site occupancy in *ubp2Δ* or *ubp3Δ* is drastically increased compared to WT. Does it suggest that under short heat stress, the increased ubiquitination in *ubp2Δ* and *ubp3Δ* is due to different pools of substrates or different lysine residues on the same proteins? Why do short and longer heat stress cause such a big difference in the site occupancy? How did the authors choose the heat-shock times? Do longer periods of heat-shock result in adaptation?

c. Along the same lines, the authors need to examine whether in Figure 4a the increased K63 ubiquitination in *ubp2Δ* and *ubp3Δ* is still Rsp5-dependent given the difference between short and longer heat stress.

Minor concerns:

1. Figure 5b, the interpretation for *rsp5 I537D* should be taken cautiously. In Kim et al 2001, they show this Rsp5 mutant affect both initial substrate conjugation and polyubiquitination of substrate proteins. Only Rsp5 auto-monoubiquitination is not affected. Figure S5f also indicates the same: Cdc19 monoubiquitination is also reduced.
2. Why are the doubling times in Fig 8b generally faster than those in Fig 8c?
3. Fig S2: labeling is incorrect
4. Any comment on the reduced Ubp2 and Ubp3 levels upon heat stress in Fig2d and 2e?

Reviewer #4 (Remarks to the Author)

In the manuscript Fang et al. describe a new function for the two ubiquitin hydrolases *ubp2* and *ubp3* in the degradation of proteins in response to heat shock and its relation to the two ubiquitin E3 ligases Rsp5 and Hul5. The proteases are disassembling K63 ubiquitin chains, which are generated by Rsp5 in order to allow the formation of K48 linked chains for degradation of the substrates.

The manuscript is well written. The experimental questions are addressed with elegant experiments. The new functions of *ubp2* and *ubp3* are shown and characterized in convincing experiments. The differential function of Rsp5 and Hul5 are nicely addressed.

Although the suggested model of Ubp2 and *ubp3* function is not in line with the experimental results. In figure 3e the authors show a SILAC based mass spectrometry experiment, comparing the differential effect of the two proteases on ubiquitin action site occupancy. Here it is shown that the same sites are unregulated in both strains, although both proteases are targeting K63 chains. Since both proteases have, according to the model, the same specificity the two proteases should complement the accumulation. In contrast the double knock-out should lead to the accumulation of the ubiquitin sites. Interestingly no specific Ubp2 or Ubp3 GG sites were identified. This might indicate some cooperation of the two proteases. E.g. a complex of the two proteases which is necessary for the function.

The manuscript should be published as is, but the suggested model should be revised to match the non complementing phenotype.

Point by point response to Reviewers' comments:

Reviewer #1

Overall, I find the experiments well conducted, the statistics sound where applied, and the conclusions are original. I think the manuscript has the potential to influence thinking in the field because E3s and DUBs are often thought to work in opposition to each other, not in coordination. This is a strong paper showing the cooperativity of two opposing enzymatic actions. I support publication of this manuscript should the authors address the specific points below.

We thank the reviewer for her/his positive comments.

*1. In Figure 1A, the authors saw considerable changes in ubiquitination levels after deletion of UBP2 and UBP3. A few other DUB deletions looked promising as well, like *ubp7Δ* and *ubp10Δ*. Did the authors test these other DUB delete strains in westerns like supplemental figure 1c-d? It might be worth having those as supplemental too, though lack of the data should NOT distract from the overall conclusions of the manuscript. This is just an interested desire by this reviewer to see the data and is NOT essential for publication.*

We verified the levels of ubiquitination in *ubp7Δ* and *ubp10Δ* strains in an independent experiment using Dotblots and found no significant difference. These results are included in Sup Fig 1b, and we did not characterize these mutants further.

2. I don't understand the labeling in Figure 1d-f. Did the authors do the experiment where they go from 45C to 25C for 5 min? Or is it from 25C to 45C for 5 min? I think it is probably the latter, but the authors should clarify here. This applies to Figure 2a-c as well. I think the authors need a better description of the experiment in the figure legends and results text to help in their shorthand in the actual figure. Maybe they could explicitly state this is the ratio of ubiquitin conjugates at 45C versus 25C normalized to wt?

As suggested, we modified the figure labeling and added more information in the figure legend for clarification. In the graphs we report the differences of normalized ubiquitination levels between heat-shocked cells (45°C, 5 min) and cells maintained at 25°C.

3. The authors should note if they performed a step to reverse the formaldehyde crosslinking in Figure 2d-e prior to the gel analysis but after the co-purification. The levels of the Ubp2- and Ubp3-TAP pulldowns are lower after crosslinking, suggesting that they did not because formaldehyde would most likely interfere in the binding of the TAP tag to the column. The results are convincing in the co-purification of Rsp5, but I don't think the general reader appreciates formaldehyde crosslinking and the effect that the lack of reversing it can have on purifications.

We did indeed heat our samples at 94°C for 30 min to reverse the cross linking after the IP. We have now added this information in the method section of the main text instead of just having it in the supplemental material. We have also added a new result panel to show that Ubp2 and 3 are not degraded after heat shock following reviewer 2's comments (Figure S2d). A portion of Ubp2 and Ubp3, as well as Rsp5 to a lower extent, is not soluble after both heat shock and cross linking in our experimental set up; that insoluble portion cannot be assessed by IP.

4. In Figure 3, why did the authors switch to 38C as the HS rather than 45C? It would be good to explain this shift in methodology. It wasn't obvious why they did.

In this case, we use an assay that was established by the Goldberg lab, in which they used 38°C (Medicherla B & Goldberg AL 2008 J Cell Biology). We previously showed that the Rsp5-dependent degradation of short-lived proteins also occurs at a higher heat shock temperature (42°C) but at a slightly lower amplitude (maybe due to compromised proteasome activity) (Fang NN, *et al*, 2014 Nature Cell Biology). We therefore prefer to use 38°C for this assay. For the other assays, we used 45°C as the increased conjugation is more extreme in comparison to lower heat shock temperatures (Fang NN, *et al*, 2014 Nature Cell Biology). We have modified the text to better highlight the fact that this assay was established by another lab and performed at 38°C.

5. In Figure 4C, was the His8-ubiquitin construct overexpressed? It would be important to note that in the results section and the figure legend. It doesn't pose a problem for the interpretation of the results, but the general reader might like to know.

It is expressed under a strong constitutive *GPD1* promotor and we added this information in the figure legend.

6. On page 9, the authors state "We co-expressed HARsp5 and Myc-tagged ubiquitin mutants containing only K48 or K63 in cells that were or not subjected to HS." I think it would be good to clarify that all other K residues were mutated to R for this experiment. That would not be clear to a general reader.

We added this info in the text on page 9.

7. I thought the in vitro experiments were great! I only wish Figure 6a had the time course in order with each K linkage. Though, others would disagree with me. The data is solid.

We thank the reviewer for her/his comment. We had unfortunately not performed a more detailed time series, although we showed in Figure S6b an earlier time point (15 min) for the experiment in Figure 6a (30min).

8. On page 12, the authors have a section titled "Absence of Ubp2 or Ubp3 reduces cell fitness upon stress". I would caution them to be careful and just say heat stress here. Or, they could say 'cytoplasmic misfolding stress'. I know they show canavanine stress (which is probably misfolding off the ribosome and cytoplasmic), but what about ethanol, osmotic, reducing, oxidative stresses to name a few? There are lots of other stresses where the phenomenon they describe might not have an influence. I think caution to not overstep their data in a conclusion is a good thing here.

We modified the title and text to specify that both deubiquitinases are important for heat stress.

9. While I find the manuscript generally easy to follow, there are a few misspellings and poor verb tense uses scattered throughout. It would be good to have a fresh eye edit the manuscript once it is revised.

Done.

Reviewer #2:

All experiments are of more than adequate technical quality and contain appropriate controls. The authors observe a highly interesting modulating role of DUBs that in combination with possible changes in the catalytic capabilities of a ubiquitin ligase causes a major re-arrangement of the ubiquitin landscape within cells in response to external stress. These findings should be of utmost interest to a broader readership. However, I have some concerns about the significance of some of the data. The nature of the physical Rsp5-Ubp2/Ubp3 interaction is unclear up to this point (see below). This point is critical for the understanding of the function of both DUBs in HS protein quality control. Furthermore, it is mandatory that the authors conduct additional experiments to investigate the putative temperature-induced switch in the Rsp5 activity in more detail. Should the authors succeed to address these points appropriately, I recommend publication of this interesting study in "Nature Communications".

We thank the reviewer for her/his encouraging comments. We have performed additional control experiments to address the two major points raised by the reviewer below. We agree that it will be important to further determine the nature of the physical interaction between Rsp5 and Ubp2 and 3, but this analysis is beyond the scope of this study (we have already 8 figures and 7 supplemental figures that contain a large body of work).

Fig. 2d,e: There is significantly less Ubp2-TAP and Ubp3-TAP detected in cell lysates after a relatively short exposure to heat for 20 min., whereas the amount of HA-Rsp5 and Pgk1 appear to be merely unchanged. This indicates that the TAP-tagged proteins

are rapidly degraded upon HS and, as a consequence, that Ubp2-TAP and Ubp3-TAP may represent substrates of Rsp5 rather than functional interaction partners in HS-exposed cells. Since an increased association of the DUBs with Rsp5 after HS is one of the proposed mechanisms to explain the function of these enzymes in promoting polyUB-K48 synthesis, it is critical to carefully address this issue. The authors should determine Ubp2-TAP and Ubp3-TAP protein levels in cells that are impaired in Rsp5 or proteasomal function. They should also monitor the amount of endogenous Ubp2 and Ubp3 that can be purified with Rsp5 under various conditions to rule out any impact of the TAP tag.

The reviewer brings up an important point. We verified that Ubp2 and Ubp3 were actually not degraded after heat shock and added this new data (Figure S2d). Instead, both Ubp2 and Ubp3, and Rsp5 to a lower extent, are less soluble following the combined heat shock and cross linking treatment.

The interaction between Ubp2 and Rsp5 has been well documented otherwise (Kee Y, et al 2005 EMBO; Lam et al PLoS ONE 2009). We agree that it will be important to further characterize the basis for the observed increased interactions after heat shock, but this is beyond the scope of this study.

Fig. 6: Additional experiments are required to allow a proper interpretation of the results shown in this figure. Fig. 6a: There are several issues to consider here. The overall synthesis of poly-ubiquitin chains increases with raising temperature. Moreover, it is unclear, whether the K48 and K63 K-only variants of ubiquitin display distinct intrinsic propensities for the incorporation into Ub chains at the employed temperatures. Thus, control ubiquitylation reactions containing an unrelated ligase (e.g. Hul5) are required to assess, which of the temperature-dependent changes in ubiquitin chain formation can be attributed to Rsp5.

Fig 6b: The specificity of the used antibodies for a certain type of ubiquitin chain is critical for the interpretation of this assay. Hence, a control experiment demonstrating the reactivity of these antisera towards defined ubiquitin chains should be included. Again, monitoring the activity of an unrelated ligase, like Hul5, could help to ascertain that the observed effects can be assigned to Rsp5.

Overall, a more reliable quantification of K63 and K48-linked poly-ubiquitin formation by Rsp5 at different temperatures should be included, e.g. by selected reaction monitoring (SRM) as shown in Fig. 4. Again, ubiquitylation reactions by an unrelated ubiquitin ligase must be included as controls.

We have added a new experiment in which we show that the increased of temperature did not induce a change of linkage specificity of Trim5, another K63-specific E3 ligase (Figure S6d). We could not use Hul5 as it mostly acts as an E4 elongating enzyme and is only active when associated to the 26S proteasome (Crosas B, *et al.*, 2006 Cell).

We also added Western blots to show that the anti-K63 and K48 antibodies were specific using K63 and K48 linked chains (Figure S4b). It should be noted that we first

showed the change of linkage specificity using single-K ubiquitin mutants (Figure 6a). We also previously showed that in presence of a substrate, we observed an increase of K48-linked chains by Western and SRM-based mass spectrometry (see below). However, a reviewer from a previous submission was critical about these results, as Rsp5 was also partially co-purified with the substrate in our experimental conditions, and we could not ascertain whether the ubiquitination was associated to the substrate or the E3 ligase. For this reason, we did not add these results in the current manuscript. Nevertheless, these experiments show that we then obtained similar results using Western or SRM analyses.

Figure for Point by Point Response.

A. Rsp5 ubiquitination of the His6-Wbp2 fragment bound to Nickel beads at the indicated temperatures with wild-type ubiquitin followed by Western blots of the eluted His6-Wbp2 fragment with anti-K63-chains and K48-chains. **B.** The levels of K48- and K63-linked ubiquitin conjugated in vitro by Rsp5 to the Wbp2 fragment at 30°C and 38°C for 30 min were quantified by SRM after elution of the His6-tagged substrate.

Reviewer #3:

[...] *While this work was rather detailed and comprehensive for both in vivo and in vitro analyses, the interpretations for some data and the conclusion about the "editing activity of DUBs" require additional discussion and clarification. Alternative models and/or more supporting evidence are needed prior to publication in Nature communications.*

We added several clarification points in the text and have slightly extended the discussion to present different possible models for Ubp2 and Ubp3 function upon heat shock stress. We have also added several additional control and complementary experiments (see below).

Major concern:

1. *The authors need to address why in Figure 1-3, they chose to use short (5min) heat-*

stress, but in Figure 4 and 5 they used a longer heat-stress condition (15-20min). The difference of the site occupancy in Figure 3e and S3 also requires more clarification.

We used 15 min heat shock in wild type and E3 mutant cells in the past (Fang NN et al., 2011 Nature cell Biology and Fang NN et al., 2014 Nature cell Biology), as ubiquitination levels were higher in comparison to earlier time points, allowing better quantitation. In these conditions free mono-ubiquitin is not yet fully depleted.

For mass spectrometry we have always increased the incubation time to 20 min, as it takes several minutes to heat up the required larger volumes of cells in our experimental settings. We have an explanatory sentence in supplemental material : “Longer HS incubation periods were required (20min vs 15min), as larger volumes of cells were used in comparison to other experiments”.

When we initially assessed the panel deubiquitinase mutants, we preferred to use a shorter time point. We had first hypothesized that the increased ubiquitination levels would only be seen in an earlier time point, as we were hoping to identify an Rsp5-antagonizing DUB that may delay degradation (which did not turn out to be the case!). Importantly, we also wanted to avoid depletion of free mono-ubiquitin. We added one sentence in the text to clarify this latest point. In Figures S1c and d, we showed that the increase of ubiquitination levels in *ubp2Δ* and *3Δ* cells was persistent after prolonged HS (10min) and in Figure 2a, 2b, and S1e we showed that levels were higher in *ubp2Δ* and *3Δ* cells after 15 min heat shock. We actually made a mistake when labeling Figure 2a and b: these assays were done after 15min heat shock (and not 5min as previously indicated). We corrected the error and sincerely apologize for the mistake. We also added new data showing that the increased in K63 chains was observed both after 5 and 15 min in absence of Ubp2 (Figure S4c).

*a. Based on Fig S3, the authors argue that *ubp2Δ* or *ubp3Δ* does not lead to ubiquitination of different pools of substrates. It also seems to me that these results further suggest that for a given substrate, no additional lysine residue is modified in *ubp2Δ* or *ubp3Δ* and therefore the increased ubiquitination levels are likely due to the attachment of longer ubiquitin chains "on the same residues". I presume this is the reason the authors put "mixed K48 and K63" conjugations in the Figure 5a editing model. If this is indeed what the authors wanted to suggest, they need to clarify this in the text.*

Correct. We think that Ubp2 or Ubp3 does not affect site occupancy. In page 8 we wrote: “Our data shows that there are few changes in site occupancies between wild-type and *ubp2Δ* or *ubp3Δ* cells after longer HS (i.e. 15-20 min), while we observed increased ubiquitination levels after similar HS treatments (Figure 2a and b, and Fig. S1e). Taken together these results suggest that the absence of either deubiquitinase likely leads to the attachment of longer ubiquitin chains on cytosolic heat-induced Rsp5 substrates”. We now also added a sentence in the discussion to reiterate this point: “After prolonged HS (15-20min), absence of one of the two deubiquitinases did not affect site occupancy on conjugated HS-induced substrates, indicating that the two

enzymes were likely involved in chain remodeling”.

b. However, the authors also show in Figure 3e that during short (5-min) heat stress, the diGly site occupancy in ubp2Δ or ubp3Δ is drastically increased compared to WT. [1] Does it suggest that under short heat stress, the increased ubiquitination in ubp2Δ and ubp3Δ is due to different pools of substrates or different lysine residues on the same proteins? [2] Why do short and longer heat stress cause such a big difference in the site occupancy? [3] How did the authors choose the heat-shock times? [4] Do longer periods of heat-shock result in adaptation?

1) The reviewer brought up an interesting point that we did not consider. After re-analyzing our data, we saw that proteins that were further ubiquitinated in absence of Ubp2 or 3 after a short heat-shock, but not identified as a heat-shock induced substrate after prolonged heat-shock, were more often associated to membranes (in comparison to proteins identified both after 5min and 20min). We have added these new results (Figure S3d). These results indicate that some (but not all) of the observed difference (between 5 and 20 min heat-shock) is due to a different pool of substrates. One possibility is that some of the “physiological” Rsp5 substrates were further conjugated prior to the heat shock in absence of the two deubiquitinases. To test this idea, we would need to properly compare within the same experiment: *UBP2* (25°C), *UBP2* (45°C, 5min), *ubp2Δ* (25°C) and *ubp2Δ* (45°C, 5min), an experiment that is technically challenging and beyond the scope of this study.

2) One reason we may only see a difference in site occupancy after short heat shock is that only 1-2 ubiquitin moieties may be added to Rsp5 substrates after that point. In this case, presence of the deubiquitinases could have a greater impact on site occupancy (after removing ubiquitin). In contrast when longer chains are established by the E3 (after potentially multiple encounters with the same substrate), removal of 1-2 ubiquitin moieties from the distal end would not influence site occupancy on substrate lysines. To properly determine whether conjugation sites were potentially affected or not by the absence of the deubiquitinases, we would have to redo the proteomic analyses and compare 0, 5 and 15min heat-shock within the same mass spectrometry analysis, ideally in both *UBP2* and *ubp2Δ* cells. This work would also require more refine *in vitro* experiments to show that the trimming occur mostly from distal ends. While we agree this is an interesting point that deserves to be further developed, we believe that it is beyond the scope of this work. The main message here is: the lasting effect of the absence of one of the two deubiquitinases after heat shock is that neither the substrate composition nor conjugation sites are altered, while chain linkage topology is affected and proteasome degradation impaired (proteasome degradation in our assays is also measured after longer heat shock periods).

3) See answer above for further clarifications of the selection of different heat-shock periods for some experiments.

4) If the acute heat-shock treatments at 42 or 45°C are extended to longer time periods (45-60min), most cells will dye. This would indicate that there isn't much adaptation in this case. Our mass spectrometry experiments showed that only a few chaperones are further expressed after heat shock at 45°C. However, translation is mostly shut down in these conditions. Nevertheless, we noticed that the increased ubiquitination upon heat-shock was not as prominent when the cells were grown at higher temperatures (e.g., 37°C) for 1-2 hours prior to more acute heat shock. In these conditions, cells express a higher levels of chaperone proteins and have therefore a higher "proteostatic" capacity that may explain the reduce levels of ubiquitination. We haven't tested cell viability in these conditions (we also didn't pre-incubate our cells at 37°C for any of the heat shock experiment presented here).

*c. Along the same lines, the authors need to examine whether in Figure 4a the increased K63 ubiquitination in *ubp2Δ* and *ubp3Δ* is still Rsp5-dependent given the difference between short and longer heat stress.*

We showed that there is no increased ubiquitination in *rsp5-1* cells in absence of Ubp2 (Figure 2a) and Ubp2 (2b) after 15 min heat shock. We verified that the employed antibody recognizes both K63 and K48 chains (see below), and therefore conclude that there is no K63-link chains that accumulate in these cells.

Figure to show reactivity of the MAB1510 anti-ubiquitin antibody (Millipore). The same K48- and K63-linked chains shown in Figure S4b (and purchased from Boston Biochem) were analyzed by Western Blot using MAB1510 (1:3000).

Minor concerns:

*1. Figure 5b, the interpretation for *rsp5 I537D* should be taken cautiously. In Kim et al 2001, they show this Rsp5 mutant affect both initial substrate conjugation and polyubiquitination of substrate proteins. Only Rsp5 auto-monoubiquitination is not affected. Figure S5f also indicates the same: Cdc19 monoubiquitination is also reduced.*

Point taken. We slightly modified the text in p10 to better caution these results. We acknowledge that mono-ubiquitination of Cdc19 appears to be affected in S5f, however quantification revealed that poly-ubiquitination is much more affected.

2. Why are the doubling times in Fig 8b generally faster than those in Fig 8c?

We found that very small variations could affect the delay especially when we performed these experiments in synthetic media. We also use strains from slightly different background and, in our synthetic media, the BY4742 cells (in 8c) grow slightly slower than BY4741 cells (in 8b).

3. Fig S2: labeling is incorrect

We corrected the error. Thank you.

4. Any comment on the reduced Ubp2 and Ubp3 levels upon heat stress in Fig2d and 2e?

We think that a portion of Ubp2 and 3, as well as Rsp5 to a lower extent, are not soluble after both heat shock and cross linking, and therefore cannot be assessed by IP. We verified that Ubp2 and 3 are not degraded after heat shock and added this new data (Figure S2d).

Reviewer #4:

The manuscript is well written. The experimental questions are addressed with elegant experiments. The new functions of ubp2 and ubp3 are shown and characterized in convincing experiments. The differential function of Rsp5 and Hul5 are nicely addressed.

We thank the reviewer for her/his positive comments.

Although the suggested model of Ubp2 and ubp3 function is not in line with the experimental results. In figure 3e the authors show a SILAC based mass spectrometry experiment, comparing the differential effect of the two proteases on ubiquitin action site occupancy. Here it is shown that the same sites are unregulated in both strains, although both proteases are targeting K63 chains. Since both proteases have, according to the model, the same specificity the two proteases should complement the accumulation. In contrast the double knock-out should lead to the accumulation of the ubiquitin sites. Interestingly no specific Ubp2 or Ubp3 GG sites were identified. This might indicate some cooperation of the two proteases. E.g. a complex of the two proteases which is necessary for the function.

The manuscript should be published as is, but the suggested model should be revised to match the non complementing phenotype.

The proposed cooperativity of the two deubiquitinases is one possible model. We have added a full paragraph in the discussion related to this point:

“Upon acute HS, both Ubp2 and Ubp3 potentially act in a cooperative manner to edit Rsp5 heat-induced misfolded substrates. Deletion of either *UBP2* or *UBP3* is sufficient to fully block degradation of misfolded proteins, and to cause the association of Rsp5 to mostly K63-linked chains. As well, largely the same substrates and ubiquitination sites were affected in the absence of either deubiquitinases in the SILAC experiments. One possibility is that both deubiquitinases are simultaneously associated to the same Rsp5-containing complex to efficiently edit ubiquitin chains on a large number of Rsp5 substrates upon heat stress. Future experiments are needed to confirm this hypothesis.”

However, it should be noted that the cooperative model is not sufficient to explain why the double deletion led to the accumulation of more conjugates (Figure 1f). In this case the double deletion could actually not lead to an additive effect as there is almost no additional free mono-ubiquitin in *ubp2Δ* cells in these conditions (5 min heat shock at 45°C) (see also Figure S1c). *In vitro*, Ubp2 with Rup1 is also sufficient to prevent the assembly of K63-linked chains (Figure 6c). An alternative model is that both Ubp2 and 3 have a redundant function. In this case, deletion of one of the two deubiquitinases leads to a dosage defect. Ubp2 has probably a more prominent function as its deletion causes a higher increase in ubiquitination. More work will be required to really distinguish between the two models. Due to space limitation, we restricted the discussion to considerations related to the cooperative model.

Reviewers' Comments:

Reviewer #1 (Remarks to the Author)

I am very satisfied with how the authors responded to my comments. I support publication in Nature Communications.

Reviewer #2 (Remarks to the Author)

Overall, I am satisfied with the author's response to most of my initial concerns and the manuscript is now suitable for publication. There still remains a minor point that to my opinion was not correctly addressed. I wondered, whether increased binding of Ubp2-TAP and Ubp3-TAP to HA-Rsp5 in heat-stressed cells might be at least in part attributed to the TAP-tags and asked to investigate this issue by studying endogenous, non-tagged proteins. The authors did not follow this point but refer to published work instead. These studies, however, only generally report on the binding of the DUBs to Rsp5 and do by no means compare the amount of bound proteins in unstressed/heat-treated cells. Intriguingly, the authors now state that heat-stress affects the solubility of the crosslinked Ubp2, Ubp3 and Rsp5 constructs, which in turn potentially questions the validity of the crosslink/IP studies. Hence, the authors should either conduct a more careful analysis of the Rsp5/DUB interactions or considerably soften their arguments on this issue in the text.

Reviewer #3 (Remarks to the Author)

Although some questions remain and probably will await future work that is beyond the scope of this study, this manuscript presents a thorough and very interesting analysis of the regulation and activities of the Rsp5 Ub ligase. I recommend this study to be published in Nature communications.

Minor comments:

The authors should consider swapping figure 3d-e with S3a-c. This would fit better with the results title on page 7 and the concluding sentence on page 8.

I think the authors should include the explanation for the site occupancy differences between 5' and 20' of heat-shock in the main text (as was included in the rebuttal letter). I think the readers would benefit from this explanation.

Point by point response to the reviewers (second submission):

Reviewer #1:

I am very satisfied with how the authors responded to my comments. I support publication in Nature Communications.

We thank this reviewer, as well as all the reviewers, for their time and comments.

Reviewer #2 (Remarks to the Author):

Overall, I am satisfied with the author's response to most of my initial concerns and the manuscript is now suitable for publication. There still remains a minor point that to my opinion was not correctly addressed. I wondered, whether increased binding of Ubp2-TAP and Ubp3-TAP to HA-Rsp5 in heat-stressed cells might be at least in part attributed to the TAP-tags and asked to investigate this issue by studying endogenous, non-tagged proteins. The authors did not follow this point but refer to published work instead. These studies, however, only generally report on the binding of the DUBs to Rsp5 and do by no means compare the amount of bound proteins in unstressed/heat-treated cells. Intriguingly, the authors now state that heat-stress affects the solubility of the crosslinked Ubp2, Ubp3 and Rsp5 constructs, which in turn potentially questions the validity of the crosslink/IP studies. Hence, the authors should either conduct a more careful analysis of the Rsp5/DUB interactions or considerably soften their arguments on this issue in the text.

We have modified the text and now state that: “These results suggest that more Ubp2 and Ubp3 interact with Rsp5 upon heat stress to process Rsp5 substrates” instead of “...strongly suggest...”. We would like to note that the fact that Ubp2, 3 and Rsp5 are less soluble after heat shock and cross linking is not novel to the revised version of the manuscript and was cited in the first version: “Although less proteins were soluble and immunoprecipitated after heat-shock in these conditions...”. We show 1) that Pgk1 does not co-IP with Ubp2 and Ubp3 after heat shock and cross-linking, 2) that Rsp5 does not co-IP with Ubp2 in absence of Rup1 in our experimental conditions, and 3) that we obtained the same results in the reciprocal experiment for both Ubp2 and Ubp3. Therefore, we believe there is enough evidence to suggest that there is an increased interaction upon heat shock.

Reviewer #3 (Remarks to the Author):

Although some questions remain and probably will await future work that is beyond the scope of this study, this manuscript presents a thorough and very interesting analysis of the regulation and activities of the Rsp5 Ub ligase. I recommend this study to be published in Nature communications.

Minor comments:

The authors should consider swapping figure 3d-e with S3a-c. This would fit better with the results title on page 7 and the concluding sentence on page 8.

Following the reviewer advise, we swapped Figure 3d-f with S3a-c.

I think the authors should include the explanation for the site occupancy differences between 5' and 20' of heat-shock in the main text (as was included in the rebuttal letter). I think the readers would benefit from this explanation.

We have indeed provided the explanations that were included in the point-by-point response letter in the manuscript (major comment 1a). We prefer not to include the comments we made in point 2 of the response 1b, as they are too speculative and because we reached the word limit. My understanding is that Nature Communications will also make these comments publicly available.